# TrkA undergoes a tetramer-to-dimer conversion to open TrkH which enables changes in membrane potential

Hanzhi Zhang[1], Yaping Pan[1], Liya Hu [1], M. Ashley Hudson[2], Katrina S. Hofstetter[2], Zhichun Xu[1], Mingqiang Rong[1], Zhao Wang [1,3], B.V. Venkataram Prasad [1], Steve W. Lockless[2]*, Wah Chiu[4]* & Ming Zhou[1]*

TrkH is a bacterial ion channel implicated in $K^+$ uptake and pH regulation. TrkH assembles with its regulatory protein, TrkA, which closes the channel when bound to ADP and opens it when bound to ATP. However, it is unknown how nucleotides control the gating of TrkH through TrkA. Here we report the structures of the TrkH-TrkA complex in the presence of ADP or ATP. TrkA forms a tetrameric ring when bound to ADP and constrains TrkH to a closed conformation. The TrkA ring splits into two TrkA dimers in the presence of ATP and releases the constraints on TrkH, resulting in an open channel conformation. Functional studies show that both the tetramer-to-dimer conversion of TrkA and the loss of constraints on TrkH are required for channel gating. In addition, deletion of TrkA in *Escherichia coli* depolarizes the cell, suggesting that the TrkH-TrkA complex couples changes in intracellular nucleotides to membrane potential.

[1] Verna and Marrs McLean Department of Biochemistry and Molecular Biology, Baylor College of Medicine, Houston, TX 77030, USA. [2] Department of Biology, Texas A&M University, College Station, TX 77843, USA. [3] CryoEM Core Center, Baylor College of Medicine, Houston, TX 77030, USA. [4] Departments of Bioengineering and of Microbiology and Immunology, Stanford University, Stanford, CA 94305, USA. *email: lockless@bio.tamu.edu; wahc@stanford.edu; mzhou@bcm.edu

TrkH, KtrB, and more distantly related KdpA belong to the superfamily of K[+] transporters (SKTs) that are found in bacteria[1], fungi[2,3], and plants[4]. In bacteria, SKT proteins include closely related TrkH[5,6] (or TrkG[7]) and KtrB[8], and a more distantly related KdpA[1], which is a component in the Kdp-ATPase system[1,9] (Supplementary Fig. 1). Bacterial SKTs are evolved from tetrameric cation channels[10–12] and have been implicated in K[+] uptake[1], pH regulation and osmoregulation[13–15], and resistance to antibiotics[16–19]. While the physiological roles of many eukaryotic cation channels have been well characterized, the roles of cation channels in bacterial physiology are much less clear. Recent studies have established a role for K[+] channels in membrane potential changes in *Bacillus subtilis*[20,21] and *Corynebacterium glutamicum*[22]. However, the precise cellular function of bacterial SKTs, some of which are not stringently selective for K[+], has not yet been determined. Our studies begin to elucidate channel function by characterizing structural changes required for nucleotide-induced channel gating and demonstrating that membrane potential changes are present in bacteria with disrupted channel gating.

TrkH or KtrB proteins assemble with their respective cytosolic partner proteins TrkA or KtrA to form a stable complex[23–25] (Supplementary Fig. 1). Structures of these complexes, TrkH-TrkA from *Vibrio parahaemolyticus*[23] and KtrB-KtrA from *Bacillus subtilis*[24] and *Vibrio alginolyticus*[25], reveal an overall highly conserved architecture for both the membrane-embedded SKT proteins and the cytosolic partner proteins. Both TrkH and KtrB form homodimers, and each protomer is composed of four homologous repeats, domains 1 to 4 (D1–D4, Supplementary Figs. 2a, b and 3a, b). Each repeat resembles a proto-typical K[+] channel subunit that has two transmembrane helices connected by a reentrance loop, or M1-P-M2. There are two long intracellular loops that connect D1 and D2 and hence the D1-D2 loop, and D3 and D4 and hence the D3-D4 loop (Supplementary Fig. 2b). D3 contains an intramembrane loop located directly beneath the selectivity filter (Supplementary Figs. 2a, b and 3a, b). TrkA assembles as a homotetrameric ring, and each protomer has two homologous domains that are similar to those found in certain ligand-gated eukaryotic and prokaryotic ion channels[26–28] (Supplementary Fig. 2c–e). These domains are commonly referred to as RCK (regulator of K[+] conductance) domains[27], and the two domains in TrkA are defined as RCK1 and RCK2[23]. Thus, a TrkA tetrameric ring is comprised of eight RCK domains. KtrA, however, has only one RCK domain, and two KtrA protomers form a stable dimer. Four dimers then assemble into a KtrA ring containing eight RCK domains[24,25] (Supplementary Fig. 3c–e).

The tetrameric TrkA ring does not have a four-fold symmetry; instead, it has a two-fold symmetry that matches that of the TrkH dimer[23] (Supplementary Fig. 2d). Each RCK domain in TrkA is composed of two subdomains defined as N1 and C1 in RCK1 and N2 and C2 in RCK2 (Supplementary Figs. 2c, d). The reason for the two-fold, rather than four-fold, symmetry is that the neighboring TrkA proteins interact using the same RCK domains, N1 to N1 and N2 to N2. As a result, the tetramer is a dimer of two dimers (Supplementary Fig. 2d) and has two different interfaces. One interface is formed by the two N1 domains, which we refer to the N1-N1 interface, and another interface is formed by the two N2 domains and hence is the N2-N2 interface. Although KtrA forms a homooctomer, the octameric ring can also assume 2-fold symmetry in certain conditions[24,29].

Patch-clamp studies showed that TrkH is an ion channel with a slight preference for K[+] over Na[+] (favoring K[+] by <2-fold), and that its open probability ($P_{open}$) is regulated by ATP and ADP through the cytosolic protein TrkA[23]. In the absence of a ligand, the TrkH-TrkA complex has a $P_{open}$ of ~0.17. The $P_{open}$ increases to ~0.87 in the presence of 5 mM ATP and decreases to ~0.02 in

5 mM ADP[23]. ATPγS (1 mM) can also activate TrkHA to a $P_{open}$ of ~0.70[23]. The TrkHA homolog, KtrB-KtrA, was shown to mediate Rb[+] uptake in a liposome-based flux assay, and ATP increases Rb[+] uptake through KtrB[24].

To better understand the mechanism by which the Trk system functions, we determined the structures of TrkH-TrkA complex from *V. parahaemalyticus* in the presence of ADP (TrkHA-ADP), ATP (TrkHA-ATP), or ATPγS (TrkHA-ATPγS). These structures provide guidance for deducing how different nucleotides may induce different conformational changes in TrkA which in turn regulates gating of the ion conduction pore in TrkH. We also found that an *Escherichia coli* strain with *trkA* deleted has a membrane potential significantly more positive than that of the wild type (wt), consistent with the hypothesis that TrkA regulates gating of TrkH.

## Results

**Overview of structures.** To investigate the physical gating of TrkH by the TrkA protein, we solved the structures of the TrkHA complex in the presence of three different nucleotides. We will first give an overview of each structure and then describe structural changes in TrkA and their impact on TrkH.

The TrkHA protein was crystallized in the presence of 10 mM ADP and the structure of TrkHA-ADP was solved by X-ray crystallography to a resolution of 3.53 Å ($R_{work}/R_{free} = 0.25/0.29$, Fig. 1a, and Supplementary Table 1). The electron density map resolves 95% of TrkH and 98% of TrkA (Methods and Supplementary Figure 4a). TrkA forms a tetrameric ring (Fig. 2a), and each TrkA has a prominent non-protein density in the N2 domain in which we modeled as an ADP (Fig. 3a). Each TrkH protomer in the current TrkHA-ADP structure makes contact with the N1 and N2 domains of TrkA that we define as HN1 and HN2 interfaces (Fig. 4a, b). In the previous TrkHA-NADH structure, only the HN2 interface is resolved (Supplementary Figs. 1 and 5a). TrkH remains a dimer and is in a closed conformation (Fig. 5a and Supplementary Fig. 1).

The structure of TrkHA-ATP was solved by cryo-electron microscopy (cryo-EM) to an overall resolution of 3.0 Å (Fig. 1b, Supplementary Fig. 6a–e, Supplementary Fig. 7a–c, and Supplementary Table 2). The TrkHA protein was frozen to the cryo-EM grids in the presence of 10 mM ATP. TrkH remains a dimer and is well resolved with a local resolution better than 3.0 Å (Supplementary Figs. 6f, 7a, c). Interestingly, TrkH is in a partially open conformation and the dimer interface is also different from that of TrkHA-ADP (Fig. 5). The density map has a lower local resolution for TrkA (Supplementary Figs. 6f, g). We suspect this is due to the conformation heterogeneity of TrkA, because the densities for TrkH were significantly improved after a focused refinement. A similar maneuver did not help improve these of TrkA (see Methods). The N2 domain of TrkA is resolved while the N1 domain is partially resolved (Supplementary Fig. 6g). We were able to dock the N1 domain into the cryo-EM density map guided by the partially resolved secondary structures and the overall shape of the domain (Supplementary Figs. 7a, c). The C1 and C2 domains of TrkA are not resolved (Supplementary Fig. 6g). In addition to conformational changes in TrkH, the TrkHA-ATP structure reveals two features in TrkA. First, in the presence of ATP, the N2-N2 interface is lost, and as a result, TrkA tetramer becomes two dimers (Fig. 2b). Second, the HN1 interface is no longer present (Fig. 4a, c).

TrkA in the TrkHA-ATP cryo-EM structure is only partially resolved, making it difficult to identify the ligand–protein interactions. As an alternative approach to visualize ligand-induced structural changes in TrkA, we solved the structure of the TrkHA complex in the presence of an ATP analog, ATPγS, to

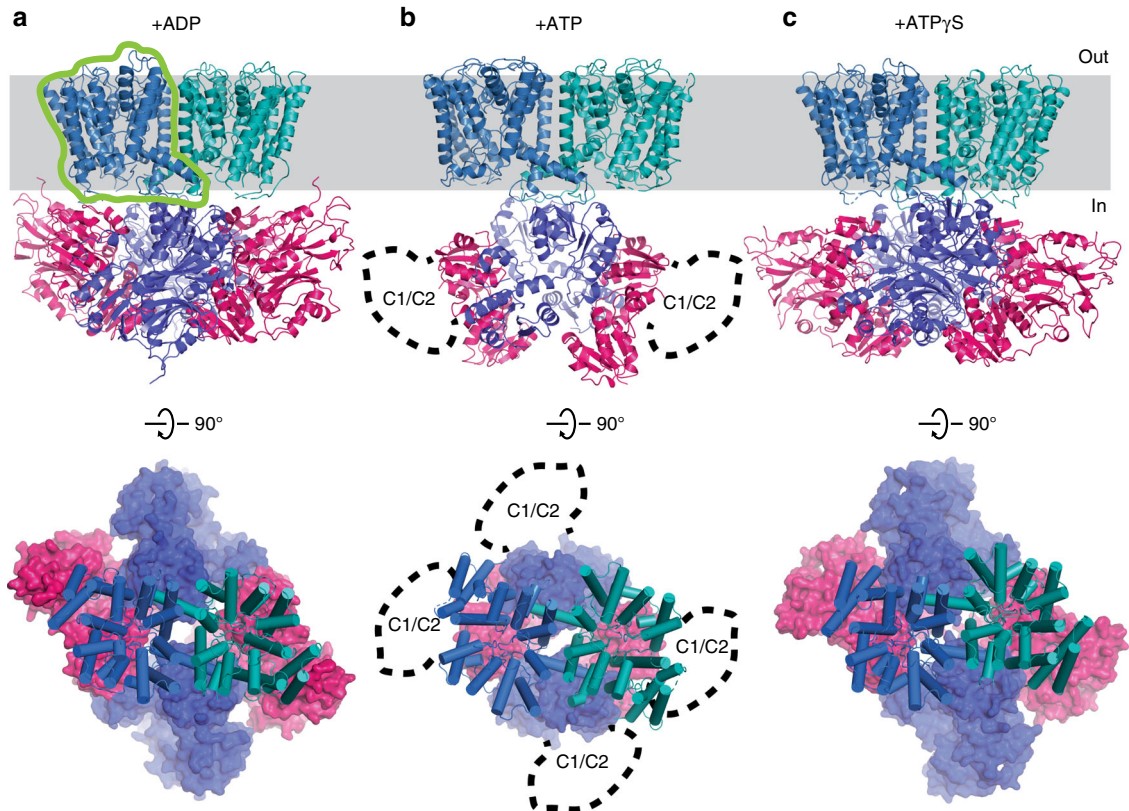

**Fig. 1 Structure of TrkHA in the presence of ADP, ATP, or ATPγS. a–c** Structures of the TrkH (blue and teal)-TrkA (dark blue and purple) in the presence of ADP (**a**), ATP (**b**), or ATPγS (**c**), viewed within the plane of the membrane (top) and from the periplasmic side (bottom). The green outline indicates an individual TrkH protomer.

3.85 Å resolution by X-ray crystallography (TrkHA-ATPγS, $R_{work}/R_{free} = 0.27/0.33$, Fig. 1c, Supplementary Fig. 4b, and Supplementary Table 1). The TrkHA protein was crystallized in the presence of 10 mM of ATPγS. The electron density map is of sufficient quality to build 98% of TrkH and 99% of TrkA (see Methods). TrkA remains a tetramer; however, the shape of the tetramer is different from that of the TrkHA-ADP (Fig. 2c). TrkH remains a dimer and is in the closed state, and while the HN2 interface is preserved, the extent of the HN1 interface is significantly reduced (Fig. 4d). As we will see in the following sections, the TrkA ring in the TrkHA-ATPγS structure assumes a conformation that is intermediate to those of TrkHA-ADP and TrkHA-ATP and allows us to deduce a likely sequence of conformational changes during gating.

**Large conformational changes in TrkA.** Major conformational changes are observed between the TrkHA-ADP and TrkHA-ATP structures. While TrkA forms a tetrameric ring in the presence of ADP, the tetramer becomes two dimers in the presence of ATP (Fig. 2a, b and Supplementary Fig. 8). In the presence of ATP, the N2 domain has a rotation of 36° relative to the N1 domain when compared to the structure of a TrkA protomer in the presence of ADP (Fig. 2d). As a result, in the presence of ATP, the N2-N2 interface is lost, while the N1-N1 interface is maintained with minor adjustments (Fig. 2a, b). In addition, because the N2 domain remains attached to the TrkH, the relative rotation between the N1 and N2 domain forces the N1 domains to move away from TrkH so that the HN1 interface is lost in the presence of ATP (Fig. 4a, c).

A similar but smaller conformational change is observed in the ATPγS-bound TrkA. The N2 domain has a 20° rotation relative to the N1 domain when compared to the structure of a TrkA in the presence of ADP (Fig. 2c, d). This is less than the rotation seen in the ATP-bound TrkA (36°), but more than that seen in a previously solved structure of TrkHA in the presence of NADH (11°) (Fig. 2d). Although relatively small, the rotation of the N2 domain in the ATPγS bound TrkA is sufficient to break the bulk of the N2-N2 interface so that there is only a tenuous connection between the two domains (Fig. 2c). Similarly, the N1 domain is pulled away from TrkH and as a result, the HN1 interface is significantly reduced (Fig. 4a, b).

**Nucleotide binding pockets in the N1 and N2 domains.** Conformation of TrkA is determined by binding of different nucleotides. In the TrkHA-ADP structure, ADP is only found in the N2 domain of TrkA (Fig. 3a), while in the TrkHA-ATPγS structure, ATPγS is found in both the N1 and N2 domains (Fig. 3b). Both the N1 and N2 domains binding to ATPγS is consistent with a previous structure of an isolated TrkA ring in the presence of ATPγS[23] (Supplementary Fig. 9 and PDB:4J9V). The ATPγS molecules in both the N1 and N2 domains assume slightly different conformations in the two structures, mainly due to rotation in the ribose and phosphate regions (Fig. 3b and Supplementary Fig. 9c). Although each TrkA protomer has similar conformation in the TrkHA-ATPγS and TrkA-ATPγS, the tetrameric TrkA ring in the TrkHA-ATPγS structure assumes a different shape than the isolated TrkA-ATPγS ring (Supplementary Fig. 9), likely because TrkH places constraints on the TrkA ring.

In the N1 binding site, conserved residues of ASN74 interact with the adenosine ring, ASN31 interacts with the ribose, and ARG98 and ARG100 form salt bridges with the γ- and β-phosphate, respectively (Fig. 3b). In the N2 domain, ATPγS and ADP interact with TrkA by an almost identical set of residues

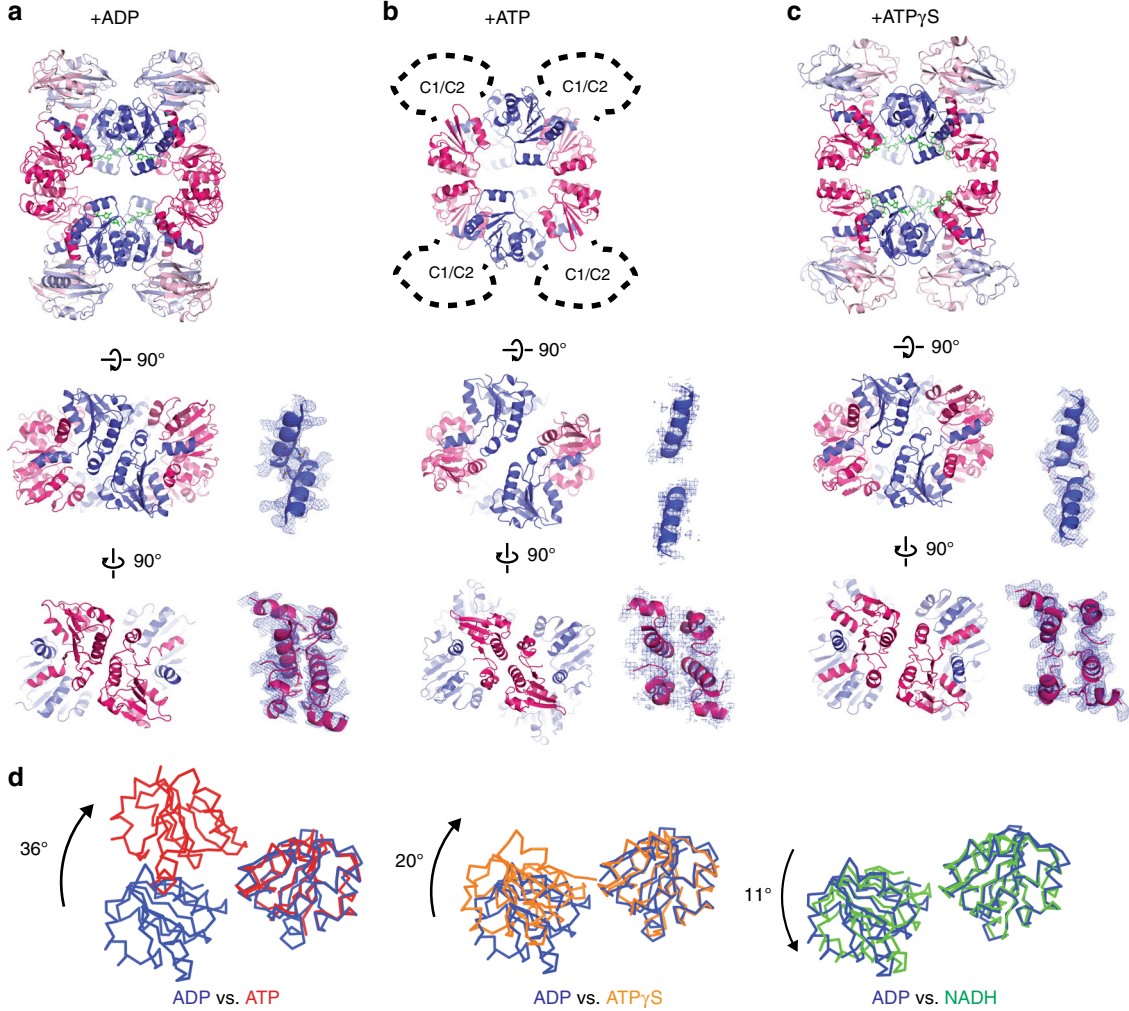

**Fig. 2 Conformational changes in TrkA monomer and tetramer. a–c** TrkA gating ring from TrkHA-ADP (**a**), TrkHA-ATPγS (**b**), and TrkHA-ATP (**c**), viewed from the periplasmic (top), N2-N2 interface (middle), and N1-N1 interface (bottom) sides. The 2Fo-Fc electron density maps of N2-N2 and N1-N1 interfaces are shown as blue mesh contoured at 1.0σ on the right. **d** Conformational changes within a TrkA protomer. N1 and N2 domains of a TrkA protomer from TrkHA-ADP (blue), TrkHA-ATPγS (orange), TrkHA-ATP (red), or TrkHA-NADH (green) are render in ribbons. The N1 domains were superimposed to show the relative rotations of the N2 domain.

(Fig. 3b). ASP283 interacts with the adenosine ring, ASN306 and ASN24 interact with the α- and β-phosphate, respectively (Fig. 3b). However, there is no coordination to the γ-phosphate. It seems that the N1 domain has conserved residues that recognize a γ-phosphate, while the N2 domain can accommodate both ADP and ATPγS. These observations led us to propose that ATP could also bind to both the N1 and N2 domains and that binding of a nucleotide to the N1 site induces conformational changes in TrkA. A previous study showed that mutating ARG100 in the N1 domain to an alanine abolishes ATP-induced channel opening, underscoring the significance of the binding site in the N1 domain[23].

**Structural changes at the TrkH/TrkA interface and in TrkH**. Structural changes in TrkA lead to changes at the TrkH-TrkA interfaces. In the TrkHA-ADP structure, each TrkH protomer makes contact with the TrkA ring in two places, the HN1 and HN2 interfaces (Fig. 4a, b). The interface is composed of interactions between TrkH THR175 and TrkA GLN40, and TrkH ARG177 and TrkA ASP44. The HN2 interface is formed by the TrkH D3-D4 loop and the TrkA N2 domain (Fig. 4a, b). The interface is formed by interactions between TrkH HIS377 and

TrkA GLU293, and TrkH ARG379 and TrkA ASP286. The HN2 interface is preserved in all the TrkHA structures (Fig. 4b–d and Supplementary Fig. 5a). Residues forming both the HN1 and HN2 interfaces are highly conserved (ConSurf[30] and Supplementary Fig. 5b).

In the TrkHA-ATP structure, the tetramer-to-dimer conversion in TrkA moves the N1 domain almost 13 Å away from TrkH. The distance between TrkH and TrkA as defined by the distance between TrkH ILE178 and TrkA ASP25 (Fig. 4a) is 11 Å in TrkHA-ADP and 24 Å in TrkHA-ATP. Movement of the N1 domains releases the HN1 interface that likely constrains the pore-lining helices to a closed state (Fig. 4a, c and 5e). The HN2 interface is maintained, but the interface moves away from the selectivity by 4 Å. The HN2 interface is 39 Å away from the selectivity filter (Fig. 4a) in the TrkHA-ATP structure, as measured between the Cαs of residues GLY222 and SER389, while the same residue pair has a distance of 35 Å in the TrkHA-ADP structure. The downward movement of the HN2 interface pulls the intramembrane loop towards the intracellular side and contributes to expansion of the permeation pathway intracellular to the selectivity filter (Fig. 5d, e). A previous study showed that deletion of this intramembrane loop results in a TrkHA that is constitutively open and no longer gated by nucleotides[23],

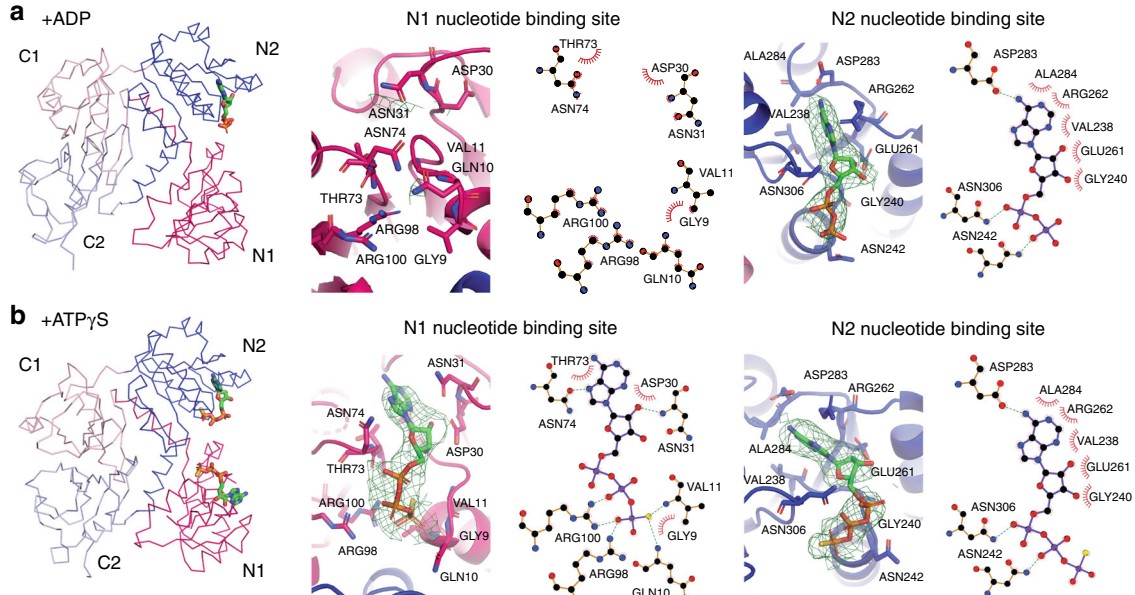

**Fig. 3 Interactions between TrkA and different nucleotides.** Ribbon representations of TrkA bound with ADP (**a**, left panel) or ATPγS (**b**, left panel), and the magnified views of the N1 (middle panels) and N2 (right panels) nucleotide binding sites. The omit map for nucleotides are shown in green meshes at 3σ. The nucleotide-protein interaction diagrams are drawn using LigPlot+[64].

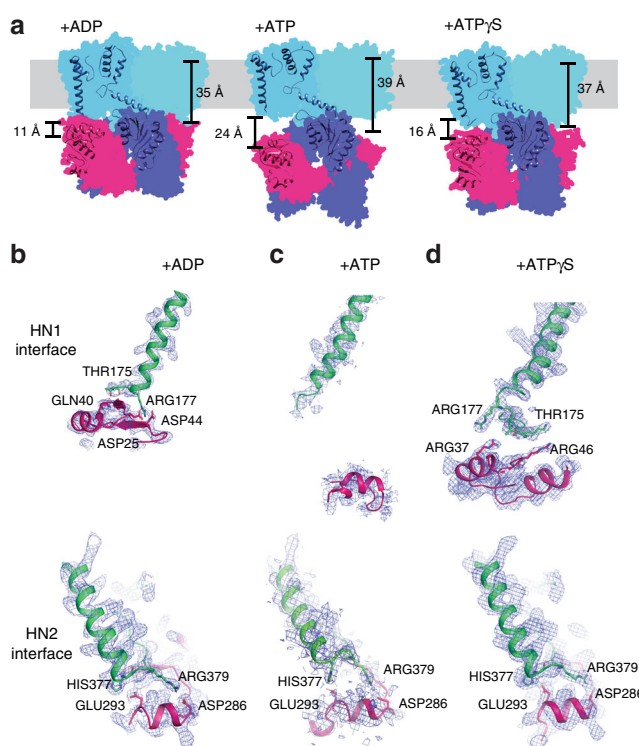

**Fig. 4 The TrkH-TrkA interfaces. a** TrkH-TrkA interactions. TrkHA-ADP (left), TrkHAγS (center), or TrkHA-ATP (right) are shown in colored schematics, viewed within the plane of the membrane. Structural elements involved in TrkH-TrkA interactions are shown as cartoon. Distances from TrkH ILE178 to TrkA ASP25 are marked on the left side of each complex. Distances from the TrkH selectivity filter to TrkH-TrkA-N2 interface (GLY222 to SER389) are marked on the right side. **b-d** Top: the HN1 interfaces from TrkHA-ADP (**a**), TrkHA-ATP (**b**), or TrkHA-ATPγS (**c**). Bottom: the HN2 interfaces from TrkHA-ADP (**a**), TrkHA-ATP (**b**), or TrkHA-ATPγS (**c**). TrkH (green) and TrkA (hot pink) are shown as cartoon. The 2Fo-Fc electron density maps are shown as blue meshes at 1.0σ cut-off.

highlighting the importance of intramembrane loop in channel gating.

Changes at the TrkH-TrkA interface leads to significant conformational changes in TrkH. TrkH from the TrkHA-ADP is in a closed state, and the ion permeation pathway is constricted with a radius of 0.9 Å at the intramembrane loop immediately below the selectivity filter (Fig. 5a, d). In contrast, TrkH in the TrkHA-ATP is in a partially open state (Fig. 5b, d) and the ion permeation pathway expands to more than 2.1 Å in radius below the selectivity filter (Fig. 5d, e). Since the selectivity filter is relatively unchanged between the ADP and ATP-bound structures, we aligned TrkH protomers by superimposing the selectivity filter. It seems that opening of the ion permeation pathway comes from an expansion of pore-lining helices and the movement of the intramembrane loop towards the intracellular side (Fig. 5e).

In addition to opening of the permeation pathway in each TrkH protomer, the TrkH dimer interface also changes. The dimer interface is mainly composed of helices D4M1 and D3M2b (Fig. 5f, g and Supplementary Fig. 2). In TrkHA-ATP, the neighboring D4M1 helices form an angle of 45°, while those of TrkHA-ADP is 59°. The change at the dimer interface is mainly due to a rigid-body rotation of the entire TrkH monomer. TrkH protomers from the TrkHA-ATPγS are in a closed state and each protomer aligns well to that from the TrkHA-ADP (Fig. 5e); however, the TrkH dimer interface is almost identical to that of the TrkH from TrkHA-ATP (Fig. 5g, h). Given that ATPγS activates the TrkHA complex almost as well as ATP, it is unclear as to why the TrkHA-ATPγS structure was not captured in an open state. Similarly, TrkH has an open probability of ~0.7 when it is not assembled with TrkA, and yet the structure of TrkH is in a closed conformation[23]. Perhaps, TrkH in a closed state is more available during the crystallization process, but it is not clear why the closed state is more favored.

**Correlation between structural changes and channel gating.** Structure comparisons suggest that conformational changes at various protein–protein interfaces are required for nucleotide-induced gating of TrkH. We applied single-channel recordings to

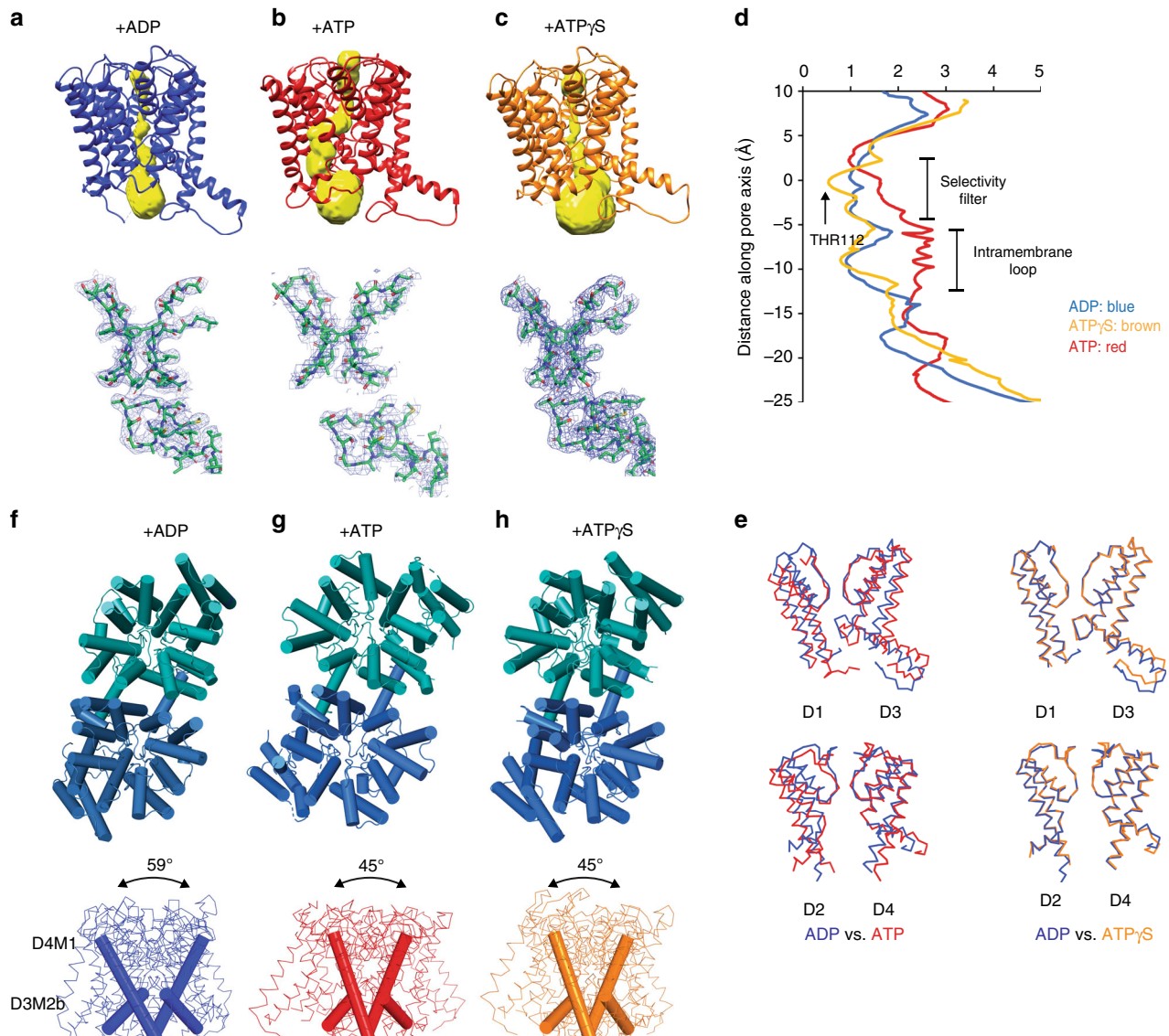

**Fig. 5 Conformational changes within TrkH. a–c** Top: Surface representations (yellow) of the ion permeation pathways of TrkH from TrkHA-ADP (**a**), TrkHA-ATPγS (**b**), or TrkHA-ATP (**c**), plotted using the program HOLE[16]. Bottom: Stick representations of the selectivity filter and intramembrane loop in TrkH from TrkHA-ADP (**a**), TrkHA-ATPγS (**b**), or TrkHA-ATP (**c**). The 2Fo-Fc electron density maps are shown in blue meshes contoured at 1.0σ. **d** Radius of the ion permeation pathways, as calculated by HOLE[16]. **e** TrkH protomers from TrkHA-ADP (blue), TrkHA-ATPγS (orange), or TrkHA-ATP (red) rendered in ribbons superimposed at the selectivity filters, showing either the D1 and D3 domains (top) or the D2 and D4 domains (bottom). **f–h** Top: TrkH protomers from TrkHA-ADP (**f**), TrkHA-ATP (**g**), TrkHA-ATPγS (**h**), rendered in cylinders viewed from the periplasmic side. Bottom: TrkH protomers from TrkHA-ADP (**f**), TrkHA-ATP (**g**), or TrkHA-ATPγS (**h**) rendered in ribbons viewed within membrane. The D4M1 and D3M2b helices at the dimer interfaces are rendered in cylinders.

test the role of the HN1 interaction in closing the channel, and the N2-N2 interface in opening the channel.

We introduced an alanine at THR175 on TrkH to perturb the HN1 interface (Fig. 4b). The TrkH (T175A) can form a stable complex with TrkA (Supplementary Fig. 10a), but its response to ADP is very different from that of the wt (Fig. 6a, c). In the presence of ADP, the $P_{open}$ for TrkHA (T175A-wt) is $0.13 \pm 0.015$ ($n = 3$), which is significantly larger than that of the wt TrkHA in ADP ($P_{open} = 0.02 \pm 0.002$, $n = 3$, $p < 0.0001$, $z$ test). In the absence of any ligand, the $P_{open}$ of TrkHA (T175A-wt) is $0.28 \pm 0.04$ ($n = 3$), which is also substantially larger than that of the TrkHA wt ($P_{open} = 0.17 \pm 0.01$, $n = 3$, $p = 0.006$, $z$ test). The increased activity is not due to altered gating by the T175A mutation because TrkH (T175A) has a $P_{open}$ of $0.66 \pm 0.02$ ($n = 3$), similar that of wt TrkH (0.67, Fig. 6b, d). In addition, TrkHA

(T175A-wt)'s response to ATP was relatively unchanged with a $P_{open}$ of $0.80 \pm 0.04$ ($n = 3$) compared to $0.87 \pm 0.003$ ($n = 3$) from the wt ($p = 0.0833$, $z$ test). These results show that perturbing the HN1 interface compromises ADP-induced channel closure, but has less effect on ATP-induced channel opening. These results are consistent with the hypothesis that HN1 interaction is required to close the channel.

Encouraged by results from the T175A, we introduced cysteine mutations to residues THR175 on TrkH and ASP40 on TrkA and examined the function of TrkHA (T175C-D40C) in reducing and oxidizing conditions. TrkHA (T175C-D40C) form a stable complex after purification (Supplementary Fig. 10a). In patch-clamp recordings, TrkHA (T175C-D40C) has a $P_{open}$ of $0.18 \pm 0.0067$ ($n = 3$), which is similar to that of the wt in the absence of any ligand ($0.17 \pm 0.01$, $n = 3$). However, in the presence of $H_2O_2$

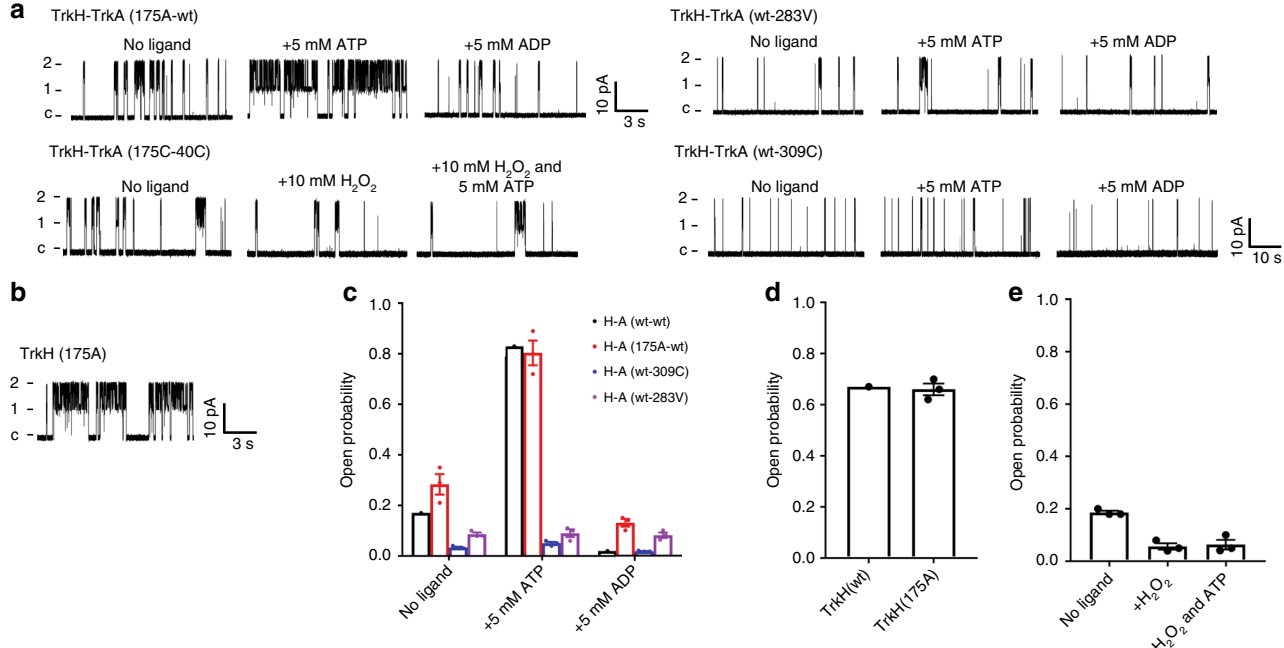

**Fig. 6 Single-channel activities of TrkHA mutants. a** Current traces of different TrkHA mutants. Top left: Current traces of TrkHA (175A-wt) in the presence of no ligand, 5 mM ATP, or 5 mM ADP. Bottom left: Current traces of TrkHAc (175C-40C), in the presence of no ligand, 10 mM $H_2O_2$, and both 10 mM $H_2O_2$ and 5 mM ATP. Top right: Current traces of TrkHA (wt-283V) in the presence of no ligand, 5 mM ATP, or 5 mM ADP. Bottom left: Current traces of TrkHA (wt-309C) in the presence of no ligand, 5 mM ATP, or 5 mM ADP. The scale bar for TrkHA (wt-309C) is 10 pA/10 s, while the scale bar for all other TrkH mutants are 10 pA/3 s. **b** Current traces of TrkH 175A (scale bar: 10 pA/3 s). **c** Open probabilities of TrkHA (wt-wt, 175A-wt, wt-309C, and wt-283V) in the presence of no ligand, 5 mM ATP, or 5 mM ADP plotted in dots overlap with bar graph. Error bars in **c–e** are standard error of the mean (s.e.m.). Charts in **c–e** were prepared in GraphPad Prism. **d** Open probabilities of TrkH (wt) and TrkH (175A) plotted in dots overlap with bar graph. **e** Open probabilities of TrkHA (175C-40C) in the presence of no ligand, 10 mM $H_2O_2$, or both 10 mM $H_2O_2$ and 5 mM ATP plotted in dots overlap with bar graph.

(10 mM), the $P_{open}$ was reduced to $0.06 \pm 0.012$ ($n = 3$, $p < 0.0001$, $z$ test, Fig. 6a, e). After the oxidation, the $P_{open}$ was unchanged even in the presence of 5 mM ATP ($0.06 \pm 0.018$, $p < 0.0001$, $z$ test, Fig. 6a, e). The effect of $H_2O_2$ can be reversed in the presence of dithiothreitol (DTT) (20 mM), in which the $P_{open}$ is recovered to 0.15. These results are consistent with the hypothesis that loss of the HN1 interaction is required to open the channel.

Although previous studies showed the importance of nucleotide binding to the N1 domain, it is not known how nucleotide binding in N2 domain may affect channel functions. To examine how binding of nucleotide to the N2 domain affect the channel activity, we introduced a valine to ASP283 at the N2 nucleotide binding site (Fig. 3b). This mutation does not affect the formation of a stable TrkHA complex (Supplementary Fig. 10a); however, it significantly affects channel activity. In the absence of any ligand, the $P_{open}$ for TrkHA (wt-D283V) was $0.08 \pm 0.006$ (Fig. 6a, c), significantly smaller than that of the wt. The addition of ADP (5 mM) no longer induces closing of the channel ($P_{open} = 0.08 \pm 0.011$), and the addition of ATP (5 mM) does not open the channel either ($P_{open} = 0.09 \pm 0.015$). It appears that loss of nucleotide binding on the N2 domain led to a TrkA ring that closes the TrkH channel regardless of the nucleotide in the N1 domain.

We also examined whether movement at the N2-N2 interface of the TrkA ring is required to open the channel. We introduced a cysteine to GLU309 at the N2-N2 interface. In the TrkHA-ADP structure, the two GLU309s from the neighboring TrkA protomers are 5.2 Å (αC-αC) away, whereas, in TrkHA-ATP, the neighboring GLU309 are 10.4 Å (αC-αC) apart (Supplementary Fig. 10a). The TrkA (E309C) mutant can assemble with TrkH to form a stable complex (Supplementary Fig. 10a). Sodium dodecyl sulfate-polyacrylamide gel electrophoresis showed that the E309C from

the neighboring TrkA protomers form a disulfide bond in the purified complex and that the disulfide bond can be reduced in the presence of 10 mM DTT (Supplementary Fig. 10a). Patch-clamp study on TrkH-TrkA (wt-E309C) showed that its $P_{open}$ is $0.035 \pm 0.0028$ ($n = 3$) with no ligand, $0.018 \pm 0.002$ ($n = 3$) in the presence of ADP, and remained low in the presence of ATP ($0.05 \pm 0.006$, $n = 3$) (Fig. 6a, c). These results suggest that mobility at the N2-N2 interface is required for TrkA-induced opening of TrkH channel. However, adding DTT during patch-clamp recordings did not restore the open probability (Supplementary Fig. 10c). We speculate that either the disulfide bridge is resistant to DTT under the patch-clamp conditions or the two cysteines enhance the N2-N2 dimer interface even in the absence of a disulfide bridge. Further studies are needed to resolve the ambiguity.

**Loss of TrkA led to depolarization of membrane potential.** TrkHA and its close relative KtrB-KtrA have been described as K⁺ transporters because bacteria with these genes deleted exhibit growth defects in low K⁺ conditions or have decreased intracellular K⁺ concentrations[31–34]. Single-channel recordings of TrkH show that it is an ion channel that has a slight preference for K⁺ over Na⁺, indicating that TrkH is able to facilitate the transport of both ions down their electrochemical gradients[23]. Since *E. coli* has a negative resting membrane potential with relatively high Na⁺ concentration outside the cell and a high K⁺ concentration inside the cell, we predict that the net ion transport will be Na⁺ into the cell upon TrkHA opening. This influx of Na⁺ would lead to a depolarized membrane.

We tested this intriguing possibility by deleting the *trkA* gene in *E. coli*, which our studies predict will lead to the opening of

TrkH channel. Using a fluorescence-based assay, the relative membrane potentials of wt *E. coli* and a *trkA* deletion strain were compared. We found that the *E. coli trkA* deletion strain was depolarized relative to wt (Fig. 7). This result is consistent with TrkH being a non-selective ion channel and suggests that the membrane potential could be altered by intracellular ATP/ADP nucleotide levels through the TrkH-TrkA complex.

## Discussion

The combined structural and functional analyses led us to propose a gating mechanism that involves a tetramer-to-dimer

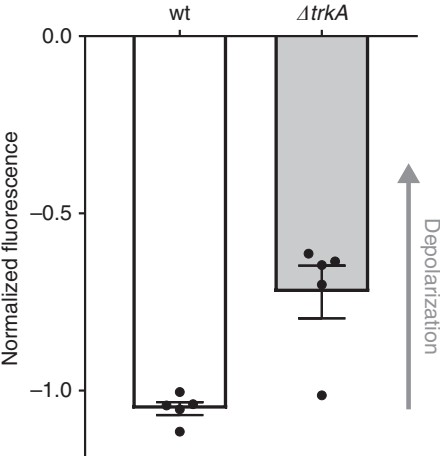

**Fig. 7 Deletion of TrkA leads to membrane potential depolarization in *E. coli*.** The relative membrane potentials of wild-type (wt) and ΔtrkA *E. coli* strains as determined by a fluorescence-based assay. The ΔtrkA cells are depolarized relative to wt cells ($n = 5$, $p < 0.001$). Points representing each measurement are overlaid with the bar graph. Error bars are s.e.m. The chart was prepared using GraphPad Prism.

conversion of TrkA and a loss of constraints on TrkH (Fig. 8, Supplementary Movie 1). ADP binds to the N2 domain and enables TrkA to form a tetrameric ring that holds TrkH shut by making contact with TrkH at both the HN1 and HN2 interfaces. These interactions, especially those at the HN1 interfaces, keep TrkH in a closed conformation. In the presence of ATP, which binds to both the N1 and N2 domains and causes a rotation of 36° between the two domains, the tetrameric TrkA gating ring splits into two TrkA dimers at the N2-N2 interface, and the conformational changes move the N1 domains away from the channel that leads to loss of the HN1 interface and allows TrkH to open. The opening of the ion permeation pathway in TrkH is achieved by the dilation of the pore-lining helices, and the movement of the intramembrane loop towards the cytoplasmic side (Fig. 5e). In addition, there is a relative rotation of the two TrkH protomers during gating (Fig. 5f, g). However, we cannot rule out that gating may require conformational changes in other regions of TrkH because TrkH in the TrkHA-ATP structure is only in a partially open state.

Structures of KtrB-KtrA complex were determined in the presence of ATP and ADP (PDB-4J7C, EMD-3450, and PDB-5BUT)[24,25,35]. The cytosolic KtrA ring interacts with the membrane-embedded KtrB in two places: the D1-D2 loop and the C terminus of KtrB (Supplementary Fig. 3b). Two mechanisms of nucleotide activation were proposed. Morais Cabral and co-workers[24,35] proposed that interaction between the D1-D2 loop and KtrA is present in the ATP-bound state and is broken in the ADP-bound state, and that the interaction is required to open the channel (Supplementary Fig. 11a). Hänelt and coworkers[25] found that the D1-D2 loop, which forms a helix in their structure, protrudes into the cytosol to interact with KtrA in the ADP-bound state, and proposed that the D1-D2 helix retracts from KtrA in the ATP-bound state to open the channel (Supplementary Fig. 11b). Since KtrB and KtrA lacks the equivalent of HN2 interface observed in TrkHA structures, it is likely that the mechanism of gating is different between KtrB-KtrA and TrkHA.

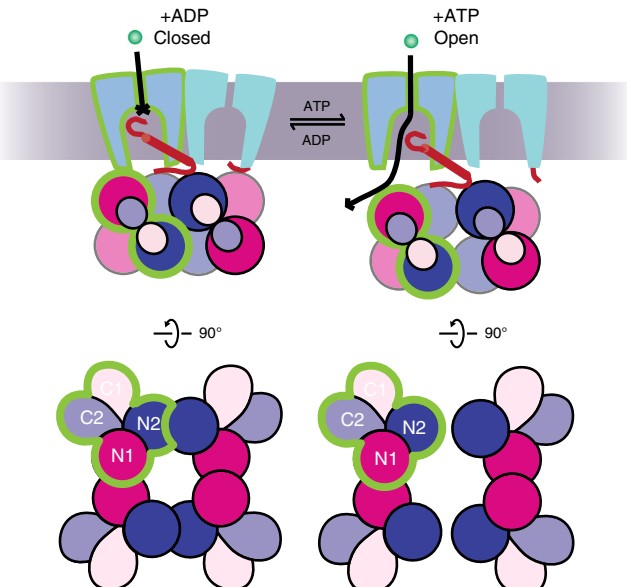

**Fig. 8 Proposed gating mechanism of TrkHA.** Upper: Cartoon illustration of TrkHA in the closed (left) and open (right) states viewed within the plane of the membrane (gray). Teal and cyan: TrkH; blue: N2 domain; purple: N1 domain; light blue: C2 domain; and light purple: C1. The green outlines mark one TrkH and one TrkA protomer. Lower: the tetrameric TrkA ring in the presence of ATP (left) and the two TrkA dimers in the presence of ATP (right). In the presence of ADP, TrkA forms a tetramer and closes TrkH through interactions at both the HN1 and HN2 interfaces. In the presence of ATP, the tetrameric TrkA gating ring is split into two dimers. Opening of a TrkH channel is achieved by the downward movement of the N2 domains away from the membrane that in turn moves the intramembrane loop from blocking the selectivity filter, and by disruption of the HN1 interfaces that allows dilation of the pore-lining helices. The two TrkH protomers also rotate relative to each other during the gating process.

More experiments are required to validate and reconcile these two mechanisms of gating in the KtrB-KtrA complex. A more recent study on a related KtrC protein, which is a close homolog to KtrA and can rescue the loss of KtrA in bacteria, showed that it also forms an octameric ring in the presence of ADP and the ring breaks up in the presence of ATP[36]. Whether breaking up of the KtrA ring is involved in KtrB gating is yet to be established.

Although the ion permeation pathway of TrkH in the ATP-bound structure is wider than those in the other structures, it may not be in a fully open state for the following reason. The narrowest region in the "open" TrkH has a radius of ~2.1 Å, which is sufficient to allow permeation of a dehydrated K⁺ ion. However, we do not think the intramembrane loop is part of the selectivity filter that contributes to dehydration of a ion during its permeation. In a previous study, we showed that cysteine-modifying reagent [2-(trimethylammonium) ethyl] methanethiosulfonate can access CYS220, which is located further inside of the narrowest region of the pore[23]. Thus, in a fully open TrkH, pore-lining helices or the intramembrane loop likely move farther apart than what we have observed to allow the permeation of hydrated K⁺ ions.

We previously reported a structure of TrkHA-NADH[23], and it is informative to compare its structure to the current TrkHA-ADP. The shape of the TrkA tetrameric ring is different in the two structures, and this is caused by a rotation of 11° between the N1 and N2 domains in each TrkA protomer (Fig. 2d). The N1-N1 interface remains unchanged, while the N2-N2 interface adjusts to accommodate the changes (Supplementary Figs. 2e and 5c). In addition, the C1 and C2 domains rotate by ~52° as a single unit (Supplementary Fig. 5d), indicating that these two domains are flexible as a unit. The dimer interface of TrkH is also different in the two structures. The two TrkH protomers in the TrkHA-NADH structure have an angle of 45°, similar to that of TrkHA-ATP or TrkHA-ATPγS (Supplementary Fig. 5e). These conformational changes highlight where structural changes could happen in the TrkHA complex, and since NADH induces a modest increase of open probability to ~0.16[23], it appears that smaller movements induced by NADH correlates with a smaller increase in the open probability.

Trk, Ktr, and other cation transport systems have been implicated in the response of bacteria to osmotic shock and pH homeostasis, which can also result in membrane potential adjustment[19,22,32–34,37]. Membrane potential in bacteria drive a number of well-known processes, such as flagellar-based motility[38], membrane protein translocation[39], small-molecule transport[40], and ATP synthesis[41]. We propose that the TrkH-TrkA system could be used to adjust one or more of these cellular processes by sensing nucleotide concentrations through TrkA. The opening of the TrkH ion channel would allow movement of ions down an electrochemical gradient to affect membrane potential. Together, these in vitro and in vivo results suggest an additional role for the Trk system in modulating bacterial membrane potential, which will be the focus of future studies.

## Methods

**Expression and purification of TrkH-TrkA complex.** Expression and purification of TrkH and TrkA follow previous protocols[23,42]. Full-length TrkH gene from *V. parahaemolyticus* was cloned into a modified pET vector with a tobacco etch virus (TEV) protease cleavage site, followed by 10-polyhistidine tag on the C terminal of the open reading frame (ORF). Full-length TrkA gene from *V. parahaemolyticus* was cloned into a pGEX vector with a glutathione transferase, followed by TEV cleavage site on the N terminal of the ORF. Each vector was transformed into *E. coli* BL21 (DE3) strain and cultured in LB medium supplemented with a proper antibiotics at 37 °C. When cell growth reaches an optical density of ~0.6 (600 nm), protein expression was induced by 0.3 mM isopropyl β-D-1-thiogalactopyranoside (IPTG) for 16 h at 20 °C. The cells were harvested and then lysed by gentle sonication in 150 mM KCl, 20 mM HEPES, and 10% glycerol, 1 mM phenylmethylsulfonyl fluoride, 2 mM β-mercaptoethanol, 5 mM MgCl, and DNAseI. To purify TrkH, the lysate was supplemented by 40 mM *n*-decyl-β-d-maltopyranoside (DM) and rocked in room

temperature for 2 h solubilize membrane proteins. The cell lysate was then cleared by centrifugation at 55,000 × *g* for 45 min, and the supernatant was loaded onto TALON Metal Affinity Resin packed in a gravity column pre-equilibrated with column buffer A (150 mM KCl, 20 mM HEPES, and 10% glycerol, 7.5 mM DM, and 2 mM BME), and subsequently washed by 10 column volume (CV) of column buffer A and 10 CV of column buffer A supplemented by 20 mM imidazole. The protein was eluted from the resin by column buffer A supplemented by 300 mM imidazole, and further purified by size-exclusion chromatography on a Superdex 200 10/300 GL column in 150 mM KCl, 20 mM HEPES, 3.5 mM DM, and 5 mM BME.

To purify TrkA, the lysate was supplemented by 5 mM BME and centrifuged at 55,000 × *g* for 45 min. Glutathione resin (Genescript) was added to the supernatant and the slurry was gently rocked at 4 °C for 2 h. The slurry was then loaded onto a gravity column, and washed with 10 CV of column buffer B (150 mM KCl, 20 mM HEPES, and 10% glycerol, and 5 mM BME). The beads was resuspended by 3 CV column buffer B. TEV protease was added to the slurry and incubated at room temperature for 1 h with gentle rocking. The slurry was loaded onto a gravity column again, and the flow-through was collected and further purified on a Superdex 200 column in 150 mM KCl, 20 mM HEPES, and 5 mM BME.

TrkH-TrkA complex was assembled by mixing TrkH and TrkA in 1.2:1 molar ratio, and further purified through size-exclusion chromatography with a Superdex 200 10/300 GL column (GE Health Sciences). For TrkH-TrkA complex in ADP, the buffer was 150 mM KCl, 20 mM HEPES, 5 mM β-mercaptoethanol, and 12 mM DM. For TrkH-TrkA complex in ATP or ATPγS, the buffer contained 150 mM KCl, 20 mM HEPES, pH 7.5, 5 mM β-mercaptoethanol, and 3.5 mM DM. Purified protein was concentrated to ~10 mg ml⁻¹ and ADP, ATP, or ATPγS was added to ~10 mM final concentration.

**Mutagenesis.** The TrkH and TrkA mutants were generated using the QuikChange method (Qiagen) with slight modifications. The primers for TrkH 175C are: CAGTGAAAGATACCAAAATGTGCCCGCGCATAGCCGAGACAGC and GCTGTCTCGGCTATGCGCGGGCACATTTTGGTATCTTTCACTG. The primers for TrkH 175A are: CAGTGAAAGATACCAAAATGGCGCCGCGCATAGCCGAGACAGC and GCTGTCTCGGCTATGCGCGGGCGCCATTTTGGTATCTTTCACTG.

The primers for TrkA40C are CCGACCGACTGCGTGAATTGTGCGACAAATACGACCTGCGTGT and ACACGCAGGTCGTATTTGTCGCACAATTCACGCAGTCGGTCGG. The primers for TrkA 283V are CACCATCGTGTTCTGTGGCGTTGCCGCAGACCAAGAACTG and CAGTTCTTGGTCTGCGGCAACGCCACAGAACACGATGGTG. The primers for TrkA 309C are CATCGCTCTCACCAACGAAGATtgcACCAACATCATGTCCGCTATG and CATAGCGGACATGATGTTGGTgcaATCTTCGTTGGTGAGAGCGATG. KOD Hot Start DNA polymerase (Merck) was used in all the mutagenesis experiments. All mutations were confirmed by sequencing the entire coding region of the proteins.

**Crystallization, data collection, and structure determination.** The TrkH-TrkA complex was concentrated to ~10 mg ml⁻¹ for crystallizations at 20 °C. TrkH-TrkA in the presence of 10 mM ADP was crystallized in magnesium acetate 100 mM, HEPES, pH 7.2, 100 mM, PEG400 26%, and 2-propanol 10%. Crystals started to show up after 24 h and grew to sizes for X-ray diffraction data collection in ~15 days.

TrkH-TrkA in the presence of 10 mM ATPγS was crystallized in MgCl₂ 50 mM, NH₄H₂PO₄ 50 mM, HEPES, pH 7.3, 100 mM, and PEG400 20%. Crystals started to appear after 7 days and grew to sizes for X-ray diffraction data collection in ~30 days.

X-ray diffraction data were collected at the beamline 24-ID-C (NE-CAT) at the Advanced Photon Source. The data were indexed, integrated, and scaled by HKL2000[43]. Phase for the TrkHA-ADP structure was solve by molecular replacement (Phaser[44]) using TrkH and the C1/C2 and N2 domains of TrkA (PDB: 4J9U)[23] as search models. A clear solution was identified with a log likelihood gain (LLG) = 19720 and a translation function Z-score (TFZ) = 111.6. For TrkHA-ATPγS, TrkH (PDB: 4J9U)[23] and TrkA (PDB: 4J9V)[23] were used as search models and the solution had LLG = 5038 and TFZ = 56.5. The structural models were then manually adjusted in Coot[45], and refined using Phenix[46] and Rossetta[47,48]. In the final model of TrkHA-ADP, residues 1, 62–65, 155–173, 484–485 from TrkH, and 1, 34–39, and 457–458 from TrkA were not resolved. In the final model of TrkHA-ATPγS, residues 4–42, 45–77, and 80–456 from TrkH, and 2–62 and 66–483 from TrkA were resolved. Pymol and UCSF Chimera were used for model display and figure preparation[49,50].

**Cryo-EM and data processing.** About 2.5 µl of 10 mg ml⁻¹ protein sample was loaded onto glow-discharged Quantifoil R2/1 400 mesh Cu grids, blotted for 8 s at 80% humidity, and plunged frozen by liquid ethane (Laika EM GP). Images were collected on 300 keV JEM3200FSC transmission electron microscope equipped with a K2 Summit detector (Gatan). Around 3000 micrographs were semi-automatically collected in super-resolution mode using the SerialEM[51] program. Each micrograph was collected in 5 frames s⁻¹ for 10 s, with a dose of 8e⁻ pixel⁻¹ s⁻¹, pixel size of 1.23 Å (super-resolution pixel size of 0.615 Å), and defocus range of 0.8–3.5 µm.

The image stacks were gain normalized and binned by 2. The beam-induced motion of the stacks was corrected by MotionCor[52]. Each image stack was summed up into two images using frames 2–50 and 2–16, respectively. These images were subsequently corrected by a contrast transfer function using CTFFIND4[53]. The

images summed from the frames 2–50 were used in the early processing steps. Around 3000 particles were manually picked to generate an initial model by EMAN2[54]. Reference-free 2D class averages were generated using the same set of particles and used as the reference template for auto-picking in RELION[55]. Auto-picking generated 248,029 particles and after removal of false-positive particles through visual examination and 2D classifications, 146,630 particles were used in further analysis. These particles were reconstituted into a 6 Å map through RELION 3D refinement[55]. The particles were separated into five classes by RELION 3D classification. Two of the classes (72,317 particles in total) that adopted similar conformations were combined, the particles were extracted from the images summed from the 2–16 frames, and fed into RELION for 3D refinement. The resolution of the map was improved to 3.7 Å. To better resolve the transmembrane domains, we applied a soft mask onto the transmembrane region, and performed a focused classification and refinement. The map was improved to a resolution of 2.97 Å. We also attempted to apply a soft mask onto the TrkA N1 region for further classification and refinement, but were not able to improve the resolution. Data processing and main results were illustrated in Supplementary Figs. 7 and 8.

Residues 30–36, 62–67, 90–93, 118–122, 154–177, 271–272, and 484–485 from TrkH were not resolved in the cryo-EM map, while TrkA was only partially resolved (residues 233–360), and the model was built in two steps. First, TrkH (PDB: 4J9U)[23] was docked into the cryo-EM map, and refined by Real Space Refinement from the Phenix software package[56]. The quality of the map was sufficient to trace 90% of the protein mass (Supplementary Figure 6f). TrkA was only partially resolved likely because of its conformational flexibility. The density of the N2 domain was of sufficient quality for the building the structural model, while the N1 domain (PDB: 4J9V) was docked into the map low-pass filtered to 6 Å, guided by the presence of three helices (residues 10–20, residues 77–89, and residues 102–109 (Supplementary Fig. 7a). C1 and C2 domains were not resolved in the cryo-EM map. Small adjustments to the model were then made manually in Coot[45]. Atomic displacement parameters were estimated using established protocol[57]. Pymol and UCSF Chimera were used for model display and figure making[49,50].

**Electrophysiology**. Single-channel patch-clamp recordings were performed on *E. coli* giant spheroplast. TrkH and TrkA were cloned into modified pET vectors with different antibiotic resistances (ampicillin resistance for TrkH and kanamycin resistance for TrkA) and the two were co-transformed into *E. coli* BL21(DE3) and cultured in the presence of both ampicillin and kanamycin. The spheroplasts were produced by the following procedure[58–60]. Cells carrying both vectors were inoculated in 5 ml of LB medium supplemented with both antibiotics and cultured in 37 °C with shaking. After the optical density reached 0.6, 0.5 mg ml$^{-1}$ cetaphalexin was added to prevent the cell division. After 2 h, 0.5 mM IPTG was added to induced protein expression. After another 2 h, the cells were collected by centrifugation (3000 × $g$) and resuspended in 0.5 ml of 0.8 M sucrose solution. Thirty microliters of Tris-HCl (1 M, pH 8.0), 24 μl of lysozyme (0.5 mg ml$^{-1}$), 6 μl DNAseI (5 mg ml$^{-1}$), and 6 μl EDTA (0.125 M) were added to digest the cell wall in room temperature. After 8 min, the digestion was stopped by 100 μl of termination solution (Tris-HCl, pH 8.0 (10 mM), MgCl$_2$ (20 mM), sucrose (0.7 M)).

Inside-out patches were pulled out from the spheroplasts plated on clean glass coverslips. The electrodes for patch clamp were pulled from glass (TW100F-4, World Precision Instruments) and polished (MP-803, Narishige Co.) to 7–10 MΩ resistance. The same buffer (150 mM KCl and 10 mM HEPES at pH 7.4) was used in both the bath and the pipette solution. Axon-200B patch-clamp amplifier (Molecular Devices Inc.) was used to amplify the analog signals. The signal was processed by a 1 kHz Bessel filter and was digitized by Digidata 1322a (Molecular Devices Inc.) at 100 μs. The data were processed by the pClampfit9 software.

**Measuring relative membrane potential**. The ΔtrkA strain was created by transducing ΔtrkA::kan from the trkA deletion strain (obtained from the Coli Genetic Stock Center) into a BW25113 background using P1vir, followed by selection on LB agar plates containing kanamycin[61]. The deletion of trkA was confirmed using PCR and whole-genome sequencing. Single colonies of *E. coli* WT and ΔtrkA strains were grown overnight in LB shaking at 37 °C and then diluted 1:1000 into fresh LB media the following morning. Cells were grown at 37 °C and harvested at an OD$_{600}$ of ~0.5 by centrifugation and resuspended to 8 × 10$^8$ cells ml$^{-1}$ in phosphate-buffered saline with 10 mM EDTA for 5 min[62]. The cells were pelleted and resuspended in assay buffer comprised of 130 mM NaCl, 60 mM Na$_2$HPO$_4$, 60 mM NaH$_2$PO$_4$, 10 mM glucose, 5 mM KCl, and 0.5 mM MgCl$_2$[63]. Following the addition of 30 μM DiOC$_2$(3), the resulting fluorescence (ex. 450 nm, em. 500, 670 nm) was analyzed using a BMG LABTECH CLARIOstar microplate reader. The red/green fluorescence ratios of WT and ΔtrkA were normalized to that of WT treated with 5 μM cyanide *m*-chlorophenylhydrazone (CCCP) or 0.5% dimethyl sulfoxide (DMSO) only ((Fluor$_{sample}$ − Fluor$_{CCCP}$)/(Fluor$_{CCCP}$ − Fluor$_{DMSO}$)).

**Reporting summary**. Further information on research design is available in the Nature Research Reporting Summary linked to this article.

## Data availability

The structure factors and model coordinates have been submitted in PDB and EMDB with deposition IDs TrkHA-ATP: PDB: 6V4J, EMDB: EMD-21041; TrkHA-ATPγS: PDB: 6V4L, and TrkHA-ADP: PDB: 6V4K. The source data underlying Figs. 5d, 6, and 7 and Supplementary Fig 10a are provided as a Source Data file. Other data are available from the corresponding authors by reasonable request.

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

## Acknowledgements

This work was supported by grants from NIH (HL086392 and DK122784 to M.Z., P41GM103832 to W.C., and GM132436 to S.W.L.), Cancer Prevention and Research Institute of Texas (R1223 to M.Z.), and Robert Welch Foundation (Q-1279 to B.V.V.P., Q-1967 to Z.W. and A-1742 to S.W.L.). This work used NE-CAT beamlines (GM124165), a Pilatus detector (RR029205), and an Eiger detector (OD021527) at the APS (DE-AC02-06CH11357). Cryo-EM work used cryo-EM Core Center at Baylor College of Medicine.

## Author contributions

M.Z. and H.Z. conceived the project. H.Z., M.R., Z.X., Y.P., M.A.H, K.S.H., L.H. and Z.W. conducted experiments. W.C. advised on cryo-EM data collection and processing. B.V.V.P. advised on X-ray crystallography data processing. S.W.L. designed experiments in *E. coli*. H.Z. and M.Z. wrote the initial draft and all authors participated in revising the manuscript.

## Competing interests

The authors declare no competing interests.
