## [Peer Review File · Nature Communications]

Reviewers' comments:

Reviewer #1 (Remarks to the Author):

Based on three new structures of the TrkAH complex in the presence of ADP, ATP and ATPgammaS, respectively, Zhang et al suggest a novel mechanism of nucleotide-dependent gating of an RCK protein-regulated K⁺ channel. This is particularly interesting as for many of those channels a mechanism of gating has been suggested, but sufficient evidence from structural and functional experiments is missing. Here, for TrkAH Zhang et al. propose that ATP binding leads to the dissociation of the tetrameric TrkA ring into two TrkA dimers, which consequently opens the pore subunits by widening the pore-lining helices and displacing the intramembrane loop. In contrast, two interaction sites of TrkA and TrkH in the ADP-bound conformation stabilize the channel in a closed conformation. This new hypothesis will be of high interest for the K⁺ channel community and the membrane transport field in general. However, to me the manuscript needs a significant improvement in data representation to justify the drawn conclusions and additional functional experiments are required to prove the raised hypothesis. Most important are the following aspects:

1. For all relevant areas (HN1 interface, HN2 interface, intramembrane loop, selectivity filter, ligands) the density maps should be represented along with the models to visualize the reliability of the models. In particular, the authors mention that residues 155 – 173 of TrkH and 34 – 39 of TrkA in the ADP-bound state were not resolved (although at least residues 37 – 39 of TrkA are shown in figure 4 and – strangely – are helical and should be better resolved than the successive loop). Hence I wonder how well the hydrogen bond between Thr175 from TrkH and Gln40 from TrkA and the salt bridge between Arg177 from TrkH and Asp44 from TrkA are resolved. A clear HN1/HN2 figure at all three conditions (similar to Fig. 4a) with density maps is required in the main manuscript. Instead, I suggest to remove Fig. 4b as it is repetition of Fig. 1a. The distances as indicated in Fig. 4b could be indicated there.
2. In support of the hypothesized role of the HN1 interface the authors should provide a comparison with the available TrkH structure. How does the TrkH structure compare to the ATP-bound TrkAH structure? One would assume that the pore-lining helices adopt similar conformations because both are not restricted. The intramembrane loop on the other hand could adopt a different conformation because the “pulling element” is missing. Or is the assembled ring rather pushing on the intramembrane loop (cf. line 157)? How do the patch clamp measurements of the T175A mutant compare to TrkH in the absence of TrkA?
3. The authors suggest that N2 binds ADP, ATP and ATPgammaS, while N1 only binds ATP and ATPgammaS (lines 190ff). If so, this finding would finally provide an explanation on how TrkAH selectively binds ATP over ADP and why under physiological conditions activation could actually happen. However, the low resolution of N1 in the ATP-bound conformation did not allow resolving the bound ligand. Further, the binding affinities of the N1 and N2 sites for the different nucleotides are not known. The manuscript lacks an explanation on why ADP would not bind to N1. I suggest the performing of ITC measurements to determine *N* values and binding affinities for the different nucleotides. The R100A mutant likely would even allow the determination of affinities for the N2 site only, while a similar mutation in the N2 site would allow the determination of N1 binding affinities. In fact, those experiments could be performed on the TrkA subunit only to further simplify the experiments.
4. Further, one should ask which role nucleotide binding to the N2 site has. Here, a mutation similar to the R100A mutation in N1 should be inserted to hinder ATP binding to the N2 site.
5. There is no evidence from any functional data for the suggested role of TrkAH (lines 297ff.). Further, to me it does not sound very plausible. Why would it be useful for the cell to pump even more protons, if the ATP concentration is already high? This in consequence would actually lead to a drastic alkalization of the cell, which most likely would be problematic. In contrast, TrkAH and KtrAB have been suggested to be involved in pH regulation. I suggest to remove this far-fetched hypothesis on the physiological role of TrkAH or to develop it better.

In addition to these five main points I have several smaller comments/questions:

6. Line 10: Explain RCK

7. Lines 116f.: It is known that at least in Gram-positive bacteria c-di-AMP binding to RCK-C closes TrkH. Thus the speculative sentence about C1 and C2 should be removed.

8. The order of the figures in the text (1,5,4,3,2,6,7) is very confusing

9. Figure S3, panel d: The resolution of the map is impossible to judge. Show a slice through the map. Panel i: Show densities of important areas (intramembrane loop, selectivity filter, AN1, AN2, ligands) and not of any random area.

10. I wonder whether you observed the dissociation of the tetrameric TrkA when adding ATP to isolated TrkA. This could be easily analysed by negative-staining EM.

11. How confident are the authors that the K⁺ is not yet rehydrated when passing the intramembrane loop (lines 147ff)? Is the opening of the pore seen really sufficient to allow K⁺ translocation

12. Which effect could the sandwich conformation have on the observed conformational changes? Might it hinder a complete dissociation of TrkA and TrkH?

13. Line 177: Specify the term "intermediate"

14. Lines 202f: "feeble HN1 interface". Show interaction with density map! Figure S6e only shows the TrkH subunit, which doesn't help the interpretation.

15. HN1 interface: What happens if you enforce the interface by a disulfide bond? Is it ATP insensitive?

16. What is the role of the relative rotation of the TrkH protomers?

17. The cartoon in Fig. 7 is very difficult to understand. Breaking of tetramer into dimers is not clear, HN1 and HN2 sites should be indicated.

18. Several panels particularly in the supplements are invisible/unreadable (e.g. I220C in S5, resolution map in S3d)

Reviewer #2 (Remarks to the Author):

The manuscript entitled "Opening of the TrkH channel by a tetramer-to-dimer conversion of the associated TrkA" describes structures of the TrkHA complex. This is a bacterial cation channel which belongs to the SKT superfamily of channels and transporters with important roles in K⁺ homeostasis of bacteria, archaea, fungi and plants. The manuscript presents three different conformational states of the TrkHA complex, providing a unique view of its mechanism of activation. Crucially, it provides the first description of the open state of a SKT channel. This manuscript will be of interest for all of those that are interested in ion channel structure and function and also for those interested in bacterial physiology.

Major issues:

1- A conclusion from this paper relates to the impact of ligand binding to the different sites (RCK1 and RCK2). In particular, ADP binds just to RCK2 while ATP γ S binds to both RCK1 and RCK2. The authors reasonably conclude that ATP will likely bind to RCK1 and RCK2. The importance of this conclusion must rely on solid data and it is surprising that the conformation of ATP γ S in RCK2 in the new structure at 3.8Å is different from the conformation of the same ligand in a previous structure at 3.0Å. These differences occur at the level of the position by the base (the base in the

new structure is flipped relative to the previous structure) and at the level of the interactions established by the phosphates (see description in lines 184 to 190 and confront with previous structure, PDB code: 4J9V). Are the authors sure that the low resolution of the new structure allows them to make these changes in the ligand? In addition, I would strongly suggest that the panels in Figure 5 show the protein structures in identical views to facilitate a comparison between the different ligands and binding sites.

2- I would like for the authors to consider that there are other structural regions that have a role in the closure/opening of TrkH besides the HN1 interface. After all their functional data shows that mutations at the interface still allow gating since there is still a large reduction in open probability (from 80% open with ATP to ~13% open with ADP versus in the WT where it changed from 87% to 2%). In addition, a) the authors describe that changes in the intramembrane loop are connected with an increase in the distance separating the selectivity filter from the HN2 interface (Lines 140 to 145). By the way, this change is not explained; and b) that the helix D3M2b helix, which is directly connected to the intramembrane loop, has moved in the ATP versus the ADP structure (Figure 2d).

Without minimizing the importance of the HN1 contact for the stability of the closed state, why are the authors ignoring all these other changes as part of the overall mechanism of moving the intramembrane loop out of the way?

3- I was surprised not to see more information about the ion pore of the "partially" open TrkHA-ATP structure. In particular, the authors have decided not to show any detailed information about the intramembrane loop (both the structure and the experimental map). It seems to me that the functional importance of this region and good resolution of the TrkHA-ATP structure merits at least a figure showing the intramembrane loop structure and the experimental map.

4- One of the major issues I have found with the manuscript is its organization. The main text does not follow the organization of the figures and this means that the data is spread-out across different figures, both main and supplementary, making reading the text and looking/thinking about the data a fairly cumbersome process.

Without being too bullish with this request, I would like to suggest that the authors consider a reorganization of the main text so that the manuscript starts with a description of the major differences between the ADP and ATP structures, followed by a description of the details that underlie those major differences and that support their mechanism proposal, including the presentation of the ATP γ S structure. I am convinced that this will increase the readability of the manuscript and its impact.

Not sure as well that the comparison of the new structures with the previous TrkHA-NADH structure should take such a prominent place in the early sections of the manuscript. It takes a whole paragraph in the section describing the TrkHA-ADP structure and also in the final paragraph of the description of TrkHA-ATP structure and becomes a distraction to the understanding of the new data. I suggest that you discuss the differences between the new structures and the TrkHA-NADH structure in a single paragraph later in the manuscript.

Minor issues:

1- The size of some of the Figures is too small. For example, all the panels in Supp. Figure 3 and Supp. Figures 4g and 4 h are virtually impossible to analyze even after extensive zooming. In addition, labels in Figures 4a, and Figure 5 are also too small.

2- Resolution of left-hand side panels in Figure 5 for ADP and ATP γ S is poor.

3- Supplementary Figure 1c serves no apparent purpose and is very difficult to analyze due to lack of contrast between ligand and protein.

4- There is something wrong with Stereo figures in Supp. Figure 4 g and h. It appears that the panels are inverted.

5- In line 114, the differences in the N2-N2 interface are mentioned when discussing changes between the TrkHA-ADP and TrkHA-NADH structures and Supp. Figure 2F is referenced. However, the figure only shows the N2-N2 interface of the TrkHA-NADH structure and so these differences are not easy to perceive.

6- Not clear what is being presented in Supp. Figures 4a and 4b, they appear to represent the two N2-N2 interfaces from the isolated TrkA-ATP γ S structure, but why do the interfaces look different?

7- The PDB code indicated in panel Supp. Figure 4e is wrong, the TrkA-ATP γ S PDB code is not 4J9U, it is 4J9V.

8- The Supp. Figure 4F panels are labeled as interfaces when they should be labeled "binding sites".

- 9- In Supp. Figure 6 the figures are labeled "Supperimposed" when it should read "Superimposed".
- 10- In Suppl. Figure 6e, the contrast between D1D2 loop residues and the rest of structure is poor, making analysis of the structure difficult.
- 11- Supplemental Table 1 should show Ramachandran plot distribution of residues as Favored, Allowed and Disallowed as this is a refinement independent parameter.

Reviewer #3 (Remarks to the Author):

The authors studied the structures of TrkH-TrkA complex in the presence of ADP or ATP. The structures of TrkHA-ADP and TrkHA-ATPS were solved by X-ray crystallography. TrkH in the TrkHA-ADP is in a closed state. The structure of TrkHA-ATP was solved by cryo-EM. TrkH in the TrkHA-ATP is in an open state. However, the N1 domain is partially resolved and the C1 and C2 domains of TrkA are not resolved at all. The major conformational change is the conversion of a tetramer to a dimer of TrkA and a loss of constraints on TrkH. Although quite a few local structures are presented, the conversion is not clear. It is better to provide a few top views of the TM region and the intracellular region separately, and label some of the dimensions. In that way, the tetramer to dimer conversion can be seen easily. Also, comparison with the published NADH bound TrkA in top view will be helpful. For local structures, it is better to label them in the top views of corresponding regions. It might be better to re-organize the figures to focus on the conversion from tetramer to dimer.

Here are some detailed comments.

- In the introduction, it is better to give more information about the difference between TrkH and KtrB, and TrkA and KtrA.
- Fig 1: Are A and B in the same orientation?
- Fig 2C: It is better to replace yellow with another color. It is easy to confuse it with the yellow color in A.
- Fig S2A: Is there any reason to show D2, D4, D1, and D3 instead of D1-4?
- Page 6: "Second, the two TrkH protomers have a relative rotation of 12.5 (Supplementary Figures 6a and b)." Are the two protomers tilted in the plane perpendicular to the membrane or rotated in the plane parallel to the membrane?
- Page 6: "The N1-N1 interface remains unchanged while the N2-N2 interface adjusts to accommodate the changes (Supplementary Figure 2f)." Fig s2f doesn't show the difference or similarity at all.
- Page 7: "In the presence of ATP, TrkAs become two dimers and no longer form a tetrameric ring (Figure 3a and b)." It is really hard to see the dimers. A top view of the TrkAs will be helpful. A comparison of the whole intracellular dimers/rings will be needed.
- Page 25: "TrkH-TrkA complex was assembled by mixing TrkH and TrkA in 1.2 to 1 molar ratio." Based on the structure, the molar ratio is 1 TrkH to 2 TrkA. Why was 1.2 to 1 molar ratio used?
- Page 27: ". Each micrograph was collected in 5 frames/s for 10s, with a dose of 8e-/Å/s". Should it be 8e-/pixel/s?
- Page 27: "These particles were reconstituted into a 6 Å map through RELION 3D refinement". Shouldn't it be reconstructed instead of reconstituted?
- Page 27: Was any symmetry used in 3D reconstruction?

Point-by-Point response to the Reviewers is highlighted in blue.

Reviewers' comments:

Reviewer #1 (Remarks to the Author):

Based on three new structures of the TrkAH complex in the presence of ADP, ATP and ATPgammaS, respectively, Zhang et al suggest a novel mechanism of nucleotide-dependent gating of an RCK protein-regulated K⁺ channel. This is particularly interesting as for many of those channels a mechanism of gating has been suggested, but sufficient evidence from structural and functional experiments is missing. Here, for TrkAH Zhang et al. propose that ATP binding leads to the dissociation of the tetrameric TrkA ring into two TrkA dimers, which consequently opens the pore subunits by widening the pore-lining helices and displacing the intramembrane loop. In contrast, two interaction sites of TrkA and TrkH in the ADP-bound conformation stabilize the channel in a closed conformation. This new hypothesis will be of high interest for the K⁺ channel community and the membrane transport field in general. However, to me the manuscript needs a significant improvement in data representation to justify the drawn conclusions and additional functional experiments are required to prove the raised hypothesis. Most important are the following aspects:

1. For all relevant areas (HN1 interface, HN2 interface, intramembrane loop, selectivity filter, ligands) the density maps should be represented along with the models to visualize the reliability of the models. In particular, the authors mention that residues 155 – 173 of TrkH and 34 – 39 of TrkA in the ADP-bound state were not resolved (although at least residues 37 – 39 of TrkA are shown in figure 4 and – strangely – are helical and should be better resolved than the successive loop). Hence I wonder how well the hydrogen bond between Thr175 from TrkH and Gln40 from TrkA and the salt bridge between Arg177 from TrkH and Asp44 from TrkA are resolved.

All the requested density maps are included in Figure 2, 4, and 5.

The side chain of ARG177 is not resolved, but the register is determined based on residues close by. To demonstrate proximity of residues ASN175 on TrkH and THR40 on TrkA, we made cysteine mutations to these two residues and we assembled the two mutations into a TrkHA complex. The cysteine pair can be crosslinked and the crosslinking mimics addition of ADP. These new data are shown in Figure 6 and Supplementary Figure 10.

A clear HN1/HN2 figure at all three conditions (similar to Fig. 4a) with density maps is required in the main manuscript. Instead, I suggest to remove Fig. 4b as it is repetition of Fig. 1a. The distances as indicated in Fig. 4b could be indicated there.

HN1/HN2 interfaces are now shown in Figure 4. Distances are marked.

2. In support of the hypothesized role of the HN1 interface the authors should provide a comparison with the available TrkH structure. How does the TrkH structure compare to the ATP-bound TrkAH structure? One would assume that the pore-lining helices adopt similar

conformations because both are not restricted. The intramembrane loop on the other hand could adopt a different conformation because the “pulling element” is missing. Or is the assembled ring rather pushing on the intramembrane loop (cf. line 157)? How do the patch clamp measurements of the T175A mutant compare to TrkH in the absence of TrkA?

The reviewer is correct that by following our logic, TrkH without constraints from TrkA should assume a more open conformation. However, the conformation we observed in TrkH alone structure is in a closed conformation that more closely resemble that of TrkH in the TrkHA-ADP. Why is TrkH in a closed state when the pore lining helices are not constrained by the TrkA ring or the intramembrane loop being pushed up by the ring? We do not have a good explanation for this other than a hand-waving argument that the closed state may have a better chance of being crystalized. The open TrkH only appeared in the cryoEM structure (TrkHA-ATP).

we measured TrkH (T175A) activity without the presence of TrkA and found that its function is similar to the wild type TrkH. The data are shown in Figure 6b.

3. The authors suggest that N2 binds ADP, ATP and ATPgammaS, while N1 only binds ATP and ATPgammaS (lines 190ff). If so, this finding would finally provide an explanation on how TrkAH selectively binds ATP over ADP and why under physiological conditions activation could actually happen. However, the low resolution of N1 in the ATP-bound conformation did not allow resolving the bound ligand. Further, the binding affinities of the N1 and N2 sites for the different nucleotides are not known. The manuscript lacks an explanation on why ADP would not bind to N1. I suggest the performing of ITC measurements to determine N values and binding affinities for the different nucleotides. The R100A mutant likely would even allow the determination of affinities for the N2 site only, while a similar mutation in the N2 site would allow the determination of N1 binding affinities. In fact, those experiments could be performed on the TrkA subunit only to further simplify the experiments.

We agree with the reviewer that obtain binding affinities of different nucleotides to the N1 and N2 domains would build a solid foundation for analysing mechanism of gating. However, we have not been able to implement ITC due to a combination of factors that include the tendency of TrkA alone to become aggregated, and hydrolysis of ATP and ATPgS that releases large amount of heat.

We are actively seeking different ways to measure nucleotide affinity and this will be the focus of more detailed studies on the gating mechanisms of TrkHA.

4. Further, one should ask which role nucleotide binding to the N2 site has. Here, a mutation similar to the R100A mutation in N1 should be inserted to hinder ATP binding to the N2 site. We agree. We made a mutation D283V in the N2 domain, and we found that it is no longer sensitive to any nucleotides. The data are shown in Figure 6a and c.

We suspect that the N2 site binds a nucleotide for structural reasons.

5. There is no evidence from any functional data for the suggested role of TrkAH (lines 297ff.). Further, to me it does not sound very plausible. Why would it be useful for the cell to pump even more protons, if the ATP concentration is already high? This in consequence would actually lead to a drastic alkalization of the cell, which most likely would be problematic. In contrast, TrkAH and KtrAB have been suggested to be involved in pH regulation. I suggest to remove this far-fetched hypothesis on the physiological role of TrkAH or to develop it better.

We speculated that opening of TrkHA would change membrane potential because it is a non-selective cation channel. We further speculated that reducing membrane depolarization would remove inhibition to proton transport. As a first step, we have measured change in membrane potential of *E. coli* in the presence and absence of TrkA. In a TrkA deletion background, the membrane potential is consistently lower when compared to wild-type *E. coli*. These data are shown in Figure 7.

Surprisingly, we found that many of the *E. coli* mutant strains obtained from large knock-out collections have additional mutations in them in addition to the intended deletion. This was discovered using whole genome sequencing of Trk system deletion strains, including those for TrkH and TrkG. We are now in the process of making clean, single gene knockouts for assessing cellular functions, which will be the focus of future studies.

In addition to these five main points I have several smaller comments/questions:

6. Line 10: Explain RCK

Done.

7. Lines 116f.: It is known that at least in Gram-positive bacteria c-di-AMP binding to RCK-C closes TrkH. Thus the speculative sentence about C1 and C2 should be removed.

Removed.

8. The order of the figures in the text (1,5,4,3,2,6,7) is very confusing

Fixed.

9. Figure S3, panel d: The resolution of the map is impossible to judge. Show a slice through the map. Panel i: Show densities of important areas (intramembrane loop, selectivity filter, AN1, AN2, ligands) and not of any random area.

Revised as instructed. See Figures 6f, 6g, and 7c.

10. I wonder whether you observed the dissociation of the tetrameric TrkA when adding ATP to isolated TrkA. This could be easily analysed by negative-staining EM.

We did the test. TrkA did not dissociate into dimers after adding ATP. The FPLC elution volume is the same. Also, in the previous TrkA-ATP γ S structure, the tetramer is "cracked" but not entirely

broken.

11. How confident are the authors that the K^+ is not yet rehydrated when passing the intramembrane loop (lines 147ff)? Is the opening of the pore seen really sufficient to allow K^+ translocation

We thank the reviewer for raising this valid point. We do not think the intramembrane loop is part of the selectivity filter that constrains an ion to the dehydrated state. In the text, we point out that the intramembrane loop in the ATP structure is only partially open and that it would likely assume a more open state so that K^+ would be hydrated when passing through the loop.

12. Which effect could the sandwich conformation have on the observed conformational changes? Might it hinder a complete dissociation of TrkA and TrkH?

This is a valid concern but we do not have a proper solution as yet. We hope that there is no “side effect” in terms of structural changes and the interactions between H and A, but we cannot be sure. To be on a more optimistic side of the argument, we showed that conclusions based on observations of the three layer structures have all been confirmed in patch-clamp studies on the 2-layer TrkHA complex.

13. Line 177: Specify the term “intermediate”

Done

14. Lines 202f: “feeble HN1 interface”. Show interaction with density map! Figure S6e only shows the TrkH subunit, which doesn’t help the interpretation.

Shown as instructed (Figure 4d).

15. HN1 interface: What happens if you enforce the interface by a disulfide bond? Is it ATP insensitive?

We made cysteine mutations to ASN175 on TrkA and THR40 on TrkH. Crosslinking the two made the channel stay closed. The results are shown in Figure 6a and e and Supplementary Figure 10b.

16. What is the role of the relative rotation of the TrkH protomers?

We are not sure, but we think that flexibility at the TrkH dimer interface is required for channel

gating because only the fully closed TrkHA-ADP structure has a 59 degree between the two interface helices. All other structures have 45 degree (Figure 5f-h).

17. The cartoon in Fig. 7 is very difficult to understand. Breaking of tetramer into dimers is not clear, HN1 and HN2 sites should be indicated.

We revised the cartoon to emphasize breaking of tetramer and loss of HN1 interface in Figure 8.

18. Several panels particularly in the supplements are invisible/unreadable (e.g. I220C in S5, resolution map in S3d)

The offending panels were deleted, and new ones were added as advised.

Reviewer #2 (Remarks to the Author):

The manuscript entitled "Opening of the TrkH channel by a tetramer-to-dimer conversion of the associated TrkA" describes structures of the TrkHA complex. This is a bacterial cation channel which belongs to the SKT superfamily of channels and transporters with important roles in K⁺ homeostasis of bacteria, archaea, fungi and plants. The manuscript presents three different conformational states of the TrkHA complex, providing a unique view of its mechanism of activation. Crucially, it provides the first description of the open state of a SKT channel. This manuscript will be of interest for all of those that are interested in ion channel structure and function and also for those interested in bacterial physiology.

Major issues:

1- A conclusion from this paper relates to the impact of ligand binding to the different sites (RCK1 and RCK2). In particular, ADP binds just to RCK2 while ATP γ S binds to both RCK1 and RCK2. The authors reasonably conclude that ATP will likely bind to RCK1 and RCK2. The importance of this conclusion must rely on solid data and it is surprising that the conformation of ATP γ S in RCK2 in the new structure at 3.8Å is different from the conformation of the same ligand in a previous structure at 3.0Å. These differences occur at the level of the position by the base (the base in the new structure is flipped relative to the previous structure) and at the level of the interactions established by the phosphates (see description in lines 184 to 190 and confront with previous structure, PDB code: 4J9V). Are the authors sure that the low resolution of the new structure allows them to make these changes in the ligand? In addition, I would strongly suggest that the panels in Figure 5 show the protein structures in identical views to facilitate a comparison between the different ligands and binding sites.

We looked into the issue of ligand conformation and feel that the one that we built is faithful to the map. This conformation is different from the previously published TrkA in isolation in the presence of ATP γ S, and we made a comment in the text.

We have changed the view to the nucleotide binding site in Figure 5 to an identical orientation. The new panels are shown in Figure 3.

2- I would like for the authors to consider that there are other structural regions that have a role in the closure/opening of TrkH besides the HN1 interface. After all their functional data shows that mutations at the interface still allow gating since there is still a large reduction in open probability (from 80% open with ATP to ~13% open with ADP versus in the WT where it changed from 87% to 2%). In addition, a) the authors describe that changes in the intramembrane loop are connected with an increase in the distance separating the selectivity filter from the HN2 interface (Lines 140 to 145). By the way, this change is not explained; and b) that the helix D3M2b helix, which is directly connected to the intramembrane loop, has moved in the ATP versus the ADP structure (Figure 2d).

Without minimizing the importance of the HN1 contact for the stability of the closed state, why are the authors ignoring all these other changes as part of the overall mechanism of moving the intramembrane loop out of the way?

We agree with the reviewer. We have added more descriptions to the movement of the intramembrane loop and the D3M2b helix. We also emphasized that the HN1 interface is part of the gating mechanism, and that other portions of the structure might be involved in gating. We stated in the Discussion that other conformational changes may exist.

3- I was surprised not to see more information about the ion pore of the "partially" open TrkHA-ATP structure. In particular, the authors have decided not to show any detailed information about the intramembrane loop (both the structure and the experimental map). It seems to me that the functional importance of this region and good resolution of the TrkHA-ATP structure merits at least a figure showing the intramembrane loop structure and the experimental map. We agree that the region mentioned by the reviewer should be highlighted. We have now shown the map and model for the selectivity filter and the intramembrane loop for all three structures in Figure 5.

4- One of the major issues I have found with the manuscript is its organization. The main text does not follow the organization of the figures and this means that the data is spread-out across different figures, both main and supplementary, making reading the text and looking/thinking about the data a fairly cumbersome process.

Without being too bullish with this request, I would like to suggest that the authors consider a reorganization of the main text so that the manuscript starts with a description of the major differences between the ADP and ATP structures, followed by a description of the details that underlie those major differences and that support their mechanism proposal, including the presentation of the ATP_γS structure. I am convinced that this will increase the readability of the manuscript and its impact.

We agree. We have rearranged the manuscript around a logical thread proposed by the Reviewer.

Not sure as well that the comparison of the new structures with the previous TrkHA-NADH structure should take such a prominent place in the early sections of the manuscript. It takes a

whole paragraph in the section describing the TrkHA-ADP structure and also in the final paragraph of the description of TrkHA-ATP structure and becomes a distraction to the understanding of the new data. I suggest that you discuss the differences between the new structures and the TrkHA-NADH structure in a single paragraph later in the manuscript.

We agree, and we revised the text as instructed.

Minor issues:

1- The size of some of the Figures is too small. For example, all the panels in Supp. Figure 3 and Supp. Figures 4g and 4 h are virtually impossible to analyze even after extensive zooming. In addition, labels in Figures 4a, and Figure 5 are also too small.

Revised as instructed.

2- Resolution of left-hand side panels in Figure 5 for ADP and ATP γ S is poor.

Revised. It was due to an error when using copy and pasted.

3- Supplementary Figure 1c serves no apparent purpose and is very difficult to analyze due to lack of contrast between ligand and protein.

Revised.

4- There is something wrong with Stereo figures in Supp. Figure 4 g and h. It appears that the panels are inverted.

Revised and stereo view confirmed (Supplementary Figure 4). The student was given a brief tutorial on the lost art of making a stereo view figure.

5- In line 114, the differences in the N2-N2 interface are mentioned when discussing changes between the TrkHA-ADP and TrkHA-NADH structures and Supp. Figure 2F is referenced. However, the figure only shows the N2-N2 interface of the TrkHA-NADH structure and so these differences are not easy to perceive.

Revised as instructed.

6- Not clear what is being presented in Supp. Figures 4a and 4b, they appear to represent the two N2-N2 interfaces from the isolated TrkA-ATP γ S structure, but why do the interfaces look different?

Revised and combined to Figure 8. The interface in TrkA-ATP γ S is different from that in TrkHA-ATP γ S.

7- The PDB code indicated in panel Supp. Figure 4e is wrong, the TrkA-ATP γ S PDB code is not 4J9U, it is 4J9V.

Revised.

8- The Supp. Figure 4F panels are labeled as interfaces when they should be labeled "binding sites".

Revised.

9- In Suppl. Figure 6 the figures are labeled “Supperimposed” when it should read “Superimposed”.

Revised.

10- In Suppl. Figure 6e, the contrast between D1D2 loop residues and the rest of structure is poor, making analysis of the structure difficult.

The panel was deleted because the D1D2 loop was not well-resolved after refinement.

11- Supplemental Table 1 should show Ramachandran plot distribution of residues as Favored, Allowed and Disallowed as this is a refinement independent parameter.

Done.

Reviewer #3 (Remarks to the Author):

The authors studied the structures of TrkH-TrkA complex in the presence of ADP or ATP. The structures of TrkHA-ADP and TrkHA-ATPS were solved by X-ray crystallography. TrkH in the TrkHA-ADP is in a closed state. The structure of TrkHA-ATP was solved by cryo-EM. TrkH in the TrkHA-ATP is in an open state. However, the N1 domain is partially resolved and the C1 and C2 domains of TrkA are not resolved at all. The major conformational change is the conversion of a tetramer to a dimer of TrkA and a loss of constraints on TrkH. Although quite a few local structures are presented, the conversion is not clear. It is better to provide a few top views of the TM region and the intracellular region separately, and label some of the dimensions. In that way, the tetramer to dimer conversion can be seen easily. Also, comparison with the published NADH bound TrkA in top view will be helpful. For local structures, it is better to label them in the top views of corresponding regions. It might be better to re-organize the figures to focus on the conversion from tetramer to dimer.

Top views are provided for both TrkH and TrkA. Comparison with the previously published TrkHA-NADH is included in Discussion. We made a better figure to clearly illustrate the TrkA tetramer to dimer conversion (Supplementary Figure 8)

Here are some detailed comments.

- In the introduction, it is better to give more information about the difference between TrkH and KtrB, and TrkA and KtrA.

Yes. We included more comparisons and emphasized the difference between TrkH and KtrB, and TrkA and KtrA.

- Fig 1: Are A and B in the same orientation?

They were in different orientations. We have revised the Figure 1 to have the same orientation.

- Fig 2C: It is better to replace yellow with another color. It is easy to confuse it with the yellow color in A.

We deleted the figure, and found a different way to describe the TrkH dimer interfaces (Figure 5f-h).

- Fig S2A: Is there any reason to show D2, D4, D1, and D3 instead of D1-4?

D1 and D3 are approximately diagonally opposed, and so are D2 and D4. Showing two at a time is for clarity (Supplementary Figure 2a and 3a).

- Page 6: "Second, the two TrkH protomers have a relative rotation of 12.5 (Supplementary Figures 6a and b)." Are the two protomers tilted in the plane perpendicular to the membrane or rotated in the plane parallel to the membrane?

The two protomers tilted in a plane perpendicular to the membrane. We now describe the movement using the interface helices as a marker. See Figure 5f-h.

- Page 6: "The N1-N1 interface remains unchanged while the N2-N2 interface adjusts to accommodate the changes (Supplementary Figure 2f)." Fig s2f doesn't show the difference or similarity at all.

The figure was revised to show the difference (Figure 2 and Supplementary Figure 8)

- Page 7: "In the presence of ATP, TrkAs become two dimers and no longer form a tetrameric ring (Figure 3a and b)." It is really hard to see the dimers. A top view of the TrkAs will be helpful. A comparison of the whole intracellular dimers/rings will be needed.

A top view is included, and a comparison of the whole intracellular ring is also included (Figure 2b and Supplementary Figure 8).

- Page 25: "TrkH-TrkA complex was assembled by mixing TrkH and TrkA in 1.2 to 1 molar ratio." Based on the structure, the molar ratio is 1 TrkH to 2 TrkA. Why was 1.2 to 1 molar ratio used? A three layer TrkHA complex is formed by 4 TrkH and 4 TrkA (Supplementary Figure 4). See also comment 12 from Reviewer 1's. A slightly higher than 1 to 1 molar is to ensure that there is sufficient amount of TrkH so that most of the complexes have three layers. This minimizes the two layer complex and reduces heterogeneity of the sample.

- Page 27: ". Each micrograph was collected in 5 frames/s for 10s, with a dose of $8e^{-}/\text{\AA}/s$ ". Should it be $8e^{-}/\text{pixel}/s$?

Yes. Corrected.

- Page 27: "These particles were reconstituted into a 6 Å map through RELION 3D refinement". Shouldn't it be reconstructed instead of reconstituted?

Yes. Corrected.

- Page 27: Was any symmetry used in 3D reconstruction?

No

REVIEWERS' COMMENTS:

Reviewer #1 (Remarks to the Author):

Zhang et al. provide a revised manuscript, which improved in readability and clarity (also about the limitations of their study). I am satisfied with most of their replies. The only remaining problem for me is the link to physiology. First, it is in no way new that potassium transporters and channels in general contribute to the establishment of the membrane potential. However, previous studies (general and on TrkHA/KtrBA) clearly showed the link between potassium accumulation and membrane potential adjustment (e.g. Kashket & Barker, 1977, Tokuda et al, 1981; Ochromel et al, 2011; Castaneda-Garcia et al, 2011), which Zhang et al. appear to question. But, their results in the changed membrane potential could as well reflect the disturbed potassium homeostasis upon abolishing the effective closure of TrkH. This said, I find the presented data with the given discussion, not further detailing what in their opinion happens other than K⁺ translocation, rather meaningless. The authors should either provide further data determining intracellular K⁺ and Na⁺ as well as the pH or remove the data. At least they should reference former papers and relate their suggestion to those publications. Also, I am not in favor of the changed title because 'TrkA alters the membrane potential' in my opinion is not even describing what the authors claim. I understand their argumentation rather as saying that the functional TrkHA system is involved in membrane potential adjustment. Additionally, there are several sentences that lack a word or have one too much as a result of changes to the manuscript (e.g. line 54., line 96, line 123, line 178). The authors should carefully revise their manuscript.

Reviewer #2 (Remarks to the Author):

I am satisfied with the changes made to the manuscript.

I only suggest that the authors revise carefully the text and figures as I found a few obvious mistakes.

For example:

Supplementary Figure 5a: The two different interfaces are both labeled as HN1. I believe that the one on the right should be labeled HN2.

Supplementary Figure 5b: One of the residues is labeled APS44, it should read ASP44.

Legend of Supplementary Figure 11. The authors have chosen to use KtrBA instead of the established KtrAB. Although it is understandable to try to harmonize the nomenclature across the field (TrkHA and KtrBA), this may cause a confusion among the readers if they try to follow the literature. I suggest that either the authors use the established KtrAB or stick to the term KtrB-KtrA used in the main text.

Point by point response to Reviewers' comments. The response is in blue.

REVIEWERS' COMMENTS:

Reviewer #1 (Remarks to the Author):

Zhang et al. provide a revised manuscript, which improved in readability and clarity (also about the limitations of their study). I am satisfied with most of their replies. The only remaining problem for me is the link to physiology. First, it is in no way new that potassium transporters and channels in general contribute to the establishment of the membrane potential. However, previous studies (general and on TrkHA/KtrBA) clearly showed the link between potassium accumulation and membrane potential adjustment (e.g. Kashket & Barker, 1977, Tokuda et al, 1981; Ochromel et al, 2011; Castaneda-Garcia et al, 2011), which Zhang et al. appear to question. But, their results in the changed membrane potential could as well reflect the disturbed potassium homeostasis upon abolishing the effective closure of TrkH. This said, I find the presented data with the given discussion, not further detailing what in their opinion happens other than K⁺ translocation, rather meaningless. The authors should either provide further data determining intracellular K⁺ and Na⁺ as well as the pH or remove the data. At least they should reference former papers and relate their suggestion to those publications.

We thank Reviewer 1 for raising this point. Our intent was to highlight the role of TrkHA/KtrBA in membrane potential adjustments based on our structural and in vitro studies rather than to question their proposed role of in potassium accumulation. From our studies, we predicted that deletion of TrkA would lead to TrkH channel opening in *E. coli* and the net flow of ions into the cell. We hypothesized that this would lead to a more depolarized membrane potential, which we experimentally tested using a fluorescence in vivo membrane potential assay. Consistent with our hypothesis, we found that the membrane potential of the TrkA deletion strain was depolarized relative to its wild-type parental strain.

We have clarified these points in the text and included the references the reviewer suggested (see reference #s 19, 32, 34):

“Bacterial SKTs are evolved from tetrameric cation channels¹⁰⁻¹² and have been implicated in K⁺ uptake¹, pH- and osmo-regulation¹³⁻¹⁵, and resistance to antibiotics¹⁶⁻¹⁹. While the physiological roles of many eukaryotic cation channels have been well characterized, the roles of cation channels in bacterial physiology are much less clear. Recent studies have established a role for K⁺ channel in membrane potential change in *Bacillus subtilis*^{20,21} and *Corynebacterium glutamicum*²².” (lines 33-39)

“TrkHA and its close relative KtrBA have been described as K⁺ transporters because bacteria with these genes deleted exhibit growth defects in low K⁺ conditions or have decreased intracellular K⁺ concentrations³¹⁻³⁴. Single-channel recordings of TrkH show that it is an ion channel that has a slight preference for K⁺ over Na⁺, indicating that TrkH is able to facilitate the transport of both ions down their electrochemical gradients²³. Since *E. coli* has a negative resting membrane potential with relatively high Na⁺ concentration outside the cell and a high K⁺ concentration inside the cell, we predict that the net ion transport will be Na⁺ into the cell upon TrkHA opening. This influx of Na⁺ would lead to a depolarized membrane.” (lines 308-316)

“Trk, Ktr and other cation transport systems have been implicated in the response of bacteria to osmotic shock and pH homeostasis, which can also result in membrane potential

adjustment^{19,22,32-34}. Membrane potential in bacteria drive a number of well-known processes, such as flagellar-based motility³⁷, membrane protein translocation³⁸, small molecule transport³⁹, and ATP synthesis⁴⁰. We propose that the TrkH-TrkA system could be used to adjust one or more of these cellular processes by sensing nucleotide concentrations through TrkA. The opening of the TrkH ion channel would allow movement of ions down an electrochemical gradient to affect membrane potential. Together, these *in vitro* and *in vivo* results suggest an additional role for the Trk system in modulating bacterial membrane potential, which will be the focus of future studies.” (lines 384-393)

Also, I am not in favor of the changed title because ‘TrkA alters the membrane potential’ in my opinion is not even describing what the authors claim. I understand their argumentation rather as saying that the functional TrkHA system is involved in membrane potential adjustment.

We thank Reviewer 1 for the clarification. We have changed the title to **“TrkA undergoes a tetramer-to-dimer conversion to open TrkH which enables changes in membrane potential”**.

Additionally, there are several sentences that lack a word or have one too much as a result of changes to the manuscript (e.g. line 54., line 96, line 123, line 178). The authors should carefully revise their manuscript.

These were fixed. Thank you.

Reviewer #2 (Remarks to the Author):

I am satisfied with the changes made to the manuscript.

I only suggest that the authors revise carefully the text and figures as I found a few obvious mistakes. For example: Supplementary Figure 5a: The two different interfaces are both labeled as HN1. I believe that the one on the right should be labeled HN2. Supplementary Figure 5b: One of the residues is labeled APS44, it should read ASP44. Legend of Supplementary Figure 11.

This was corrected. Thank you.

The authors have chosen to use KtrBA instead of the established KtrAB. Although it is understandable to try to harmonize the nomenclature across the field (TrkHA and KtrBA), this may cause a confusion among the readers if they try to follow the literature. I suggest that either the authors use the established KtrAB or stick to the term KtrB-KtrA used in the main text.

We changed KtrBA to KtrB-KtrA.